# Improving robustness against common corruptions by covariate shift adaptation

**Steffen Schneider**\*
University of Tübingen &
IMPRS-IS

**Evgenia Rusak**\*
University of Tübingen &
IMPRS-IS

**Luisa Eck**
LMU Munich

**Oliver Bringmann**[†]
University of Tübingen

**Wieland Brendel**[†]
University of Tübingen

**Matthias Bethge**[†]
University of Tübingen

## Abstract

Today's state-of-the-art machine vision models are vulnerable to image corruptions like blurring or compression artefacts, limiting their performance in many real-world applications. We here argue that popular benchmarks to measure model robustness against common corruptions (like ImageNet-C) underestimate model robustness in many (but not all) application scenarios. The key insight is that in many scenarios, multiple unlabeled examples of the corruptions are available and can be used for unsupervised online adaptation. Replacing the activation statistics estimated by batch normalization on the training set with the statistics of the corrupted images consistently improves the robustness across 25 different popular computer vision models. Using the corrected statistics, ResNet-50 reaches 62.2% mCE on ImageNet-C compared to 76.7% without adaptation. With the more robust DeepAugment+AugMix model, we improve the state of the art achieved by a ResNet50 model up to date from 53.6% mCE to 45.4% mCE. Even adapting to a single sample improves robustness for the ResNet-50 and AugMix models, and 32 samples are sufficient to improve the current state of the art for a ResNet-50 architecture. We argue that results with adapted statistics should be included whenever reporting scores in corruption benchmarks and other out-of-distribution generalization settings.

## 1 Introduction

Deep neural networks (DNNs) are known to perform well in the independent and identically distributed (*i.i.d.*) setting when the test and training data are sampled from the same distribution. However, for many applications this assumption does not hold. In medical imaging, X-ray images or histology slides will differ from the training data if different acquisition systems are being used. In quality assessment, the images might differ from the training data if lighting conditions change or if dirt particles accumulate on the camera. Autonomous cars may face rare weather conditions like sandstorms or big hailstones. While human vision is quite robust to those deviations [1], modern machine vision models are often sensitive to such image corruptions.

We argue that current evaluations of model robustness underestimate performance in many (but not all) real-world scenarios. So far, popular image corruption benchmarks like ImageNet-C [IN-C; 2] focus only on ad hoc scenarios in which the tested model has zero prior knowledge about the corruptions it encounters during test time, even if it encounters the same corruption multiple times. In the example of medical images or quality assurance, the image corruptions do not change from sample to sample

but are continuously present over a potentially large number of samples. Similarly, autonomous cars will face the same weather condition over a continuous stream of inputs during the same sand- or hailstorm. These (unlabeled) observations can allow recognition models to adapt to the change in the input distribution.

Such unsupervised adaptation mechanisms are studied in the field of domain adaptation (DA), which is concerned with adapting models trained on one domain (the source, here clean images) to another for which only unlabeled samples exist (the target, here the corrupted images). Tools and methods from domain adaptation are thus directly applicable to increase model robustness against common corruptions, but so far no results on popular benchmarks have been reported. The overall goal of this work is to encourage stronger interactions between the currently disjoint fields of domain adaptation and robustness towards common corruptions.

We here focus on one popular technique in DA, namely adapting batch normalization [BN; 3] statistics [4–6]. In computer vision, BN is a popular technique for speeding up training and is present in almost all current state-of-the-art image recognition models. BN estimates the statistics of activations for the training dataset and uses them to normalize intermediate activations in the network.

By design, activation statistics obtained during training time do not reflect the statistics of the test distribution when testing in out-of-distribution settings like corrupted images. We investigate and corroborate the hypothesis that high-level distributional shifts from clean to corrupted images largely manifest themselves in a difference of first and second order moments in the internal representations of a deep network, which can be mitigated by adapting BN statistics, i.e. by estimating the BN statistics on the corrupted images. We demonstrate that this simple adaptation alone can greatly increase recognition performance on corrupted images.

Our contributions can be summarized as follows:

- We suggest to augment current benchmarks for common corruptions with two additional performance metrics that measure robustness after partial and full unsupervised adaptation to the corrupted images.
- We draw connections to domain adaptation and show that even adapting to a single corrupted sample improves the baseline performance of a ResNet-50 model trained on IN from 76.7% mCE to 71.4%. Robustness increases with more samples for adaptation and converges to a mCE of 62.2%.
- We show that the robustness of a variety of vanilla models trained on ImageNet [IN; 7, 8] substantially increases after adaptation, sometimes approaching the current state-of-the-art performance on IN-C without adaptation.
- Similarly, we show that the robustness of state-of-the-art ResNet-50 models on IN-C consistently increases when adapted statistics are used. We surpass the best non-adapted model (52.3% mCE) by almost 7% points.
- We show results on several popular image datasets and discuss both the generality and limitations of our approach.
- We demonstrate that the performance degradation of a non-adapted model can be well predicted from the Wasserstein distance between the source and target statistics. We propose a simple theoretical model for bounding the Wasserstein distance based on the adaptation parameters.

## 2  Measuring robustness against common corruptions

The ImageNet-C benchmark [2] consists of 15 test corruptions and four hold-out corruptions which are applied with five different severity levels to the $50\,000$ test images of the ILSVRC2012 subset of ImageNet [8]. During evaluation, model responses are assumed to be conditioned only on single samples, and are not allowed to adapt to e.g. a batch of samples from the same corruption. We call this the ad hoc or non-adaptive scenario. The main performance metric on IN-C is the mean corruption error (mCE), which is obtained by normalizing the model's top-1 errors with the top-1 errors of AlexNet [9] across the $C = 15$ test corruptions and $S = 5$ severities (cf. 2):

$$\mathrm{mCE(model)} = \frac{1}{C} \sum_{c=1}^{C} \frac{\sum_{s=1}^{S} \mathrm{err}_{c,s}^{\mathrm{model}}}{\sum_{s=1}^{S} \mathrm{err}_{c,s}^{\mathrm{AlexNet}}}. \tag{1}$$

Note that mCE reflects only one possible averaging scheme over the IN-C corruption types. We additionally report the overall top-1 accuracies and report results for all individual corruptions in the supplementary material and the project repository.

In many application scenarios, this ad hoc evaluation is too restrictive. Instead, often many unlabeled samples with similar corruptions are available, which can allow models to adapt to the shifted data distribution. To reflect such scenarios, we propose to also benchmark the robustness of adapted models. To this end, we split the $50\,000$ validation samples with the same corruption and severity into batches with $n$ samples each and allow the model to condition its responses on the complete batch of images. We then compute mCE and top-1 accuracy in the usual way.

We consider three scenarios: In the *ad hoc* scenario, we set $n = 1$ which is the typically considered setting. In the *full adaptation* scenario, we set $n = 50\,000$, meaning the model may adapt to the full set of unlabeled samples with the same corruption type before evaluation. In the *partial adaptation* scenario, we set $n = 8$ to test how efficiently models can adapt to a relatively small number of unlabeled samples.

## 3   Correcting Batch Normalization statistics as a strong baseline for reducing covariate shift induced by common corruptions

We propose to use a well-known tool from domain adaptation—adapting batch normalization statistics [5, 6]—as a simple baseline to increase robustness against image corruptions in the adaptive evaluation scenarios. IN trained models typically make use of batch normalization [BN; 3] for faster convergence and improved stability during training. Within a BN layer, first and second order statistics $\mu_c, \sigma_c^2$ of the activation tensors $\mathbf{z}_c$ are estimated across the spatial dimensions and samples for each feature map $c$. The activations are then normalized by subtracting the mean $\mu_c$ and dividing by $\sigma_c^2$. During training, $\mu_c$ and $\sigma_c^2$ are estimated *per batch*. During evaluation, $\mu_c$ and $\sigma_c^2$ are estimated *over the whole training dataset*, typically using exponential averaging [10].

Using the BN statistics obtained during training for testing makes the model decisions deterministic but is also problematic if the input distribution changes. If the activation statistics $\mu_c, \sigma_c^2$ change for samples from the test domain, then the activations of feature map $c$ are no longer normalized to zero mean and unit variance, breaking a crucial assumption that all downstream layers depend on. Mathematically, this *covariate shift*[2] can be formalized as follows:

**Definition 1** (Covariate Shift, cf. 12, 13)**.** *There exists covariate shift between a source distribution with density $p_s : \mathcal{X} \times \mathcal{Y} \to \mathbb{R}^+$ and a target distribution with density $p_t : \mathcal{X} \times \mathcal{Y} \to \mathbb{R}^+$, written as $p_s(\mathbf{x}, y) = p_s(\mathbf{x})p_s(y|\mathbf{x})$ and $p_t(\mathbf{x}, y) = p_t(\mathbf{x})p_t(y|\mathbf{x})$, if $p_s(y|\mathbf{x}) = p_t(y|\mathbf{x})$ and $p_s(\mathbf{x}) \neq p_t(\mathbf{x})$ where $y \in \mathcal{Y}$ denotes the class label.*

**Removal of covariate shift.**    If covariate shift (Def. 1) only causes differences in the first and second order moments of the feature activations $\mathbf{z} = f(\mathbf{x})$, it can be removed by applying normalization:

$$p\left(\frac{f(\mathbf{x}) - \mathbb{E}_s[f(\mathbf{x})]}{\sqrt{\mathbb{V}_s[f(\mathbf{x})]}}\bigg|\mathbf{x}\right) p_s(\mathbf{x}) \approx p\left(\frac{f(\mathbf{x}) - \mathbb{E}_t[f(\mathbf{x})]}{\sqrt{\mathbb{V}_t[f(\mathbf{x})]}}\bigg|\mathbf{x}\right) p_t(\mathbf{x}). \tag{2}$$

Reducing the covariate shift in models with batch normalization is particularly straightforward: it suffices to estimate the BN statistics $\mu_t, \sigma_t^2$ on (unlabeled) samples from the test data available for adaptation. If the number of available samples $n$ is too small, the estimated statistics would be too unreliable. We therefore leverage the statistics $\mu_s, \sigma_s^2$ already computed on the training dataset as a prior and infer the test statistics for each test batch as follows,

$$\bar{\mu} = \frac{N}{N+n}\mu_s + \frac{n}{N+n}\mu_t, \quad \bar{\sigma}^2 = \frac{N}{N+n}\sigma_s^2 + \frac{n}{N+n}\sigma_t^2. \tag{3}$$

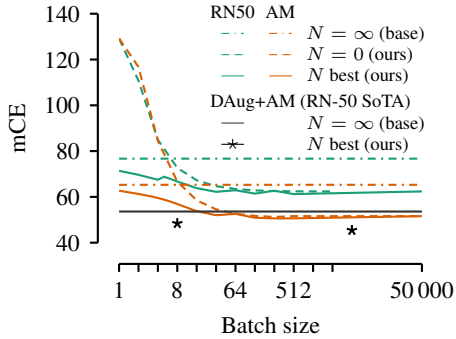
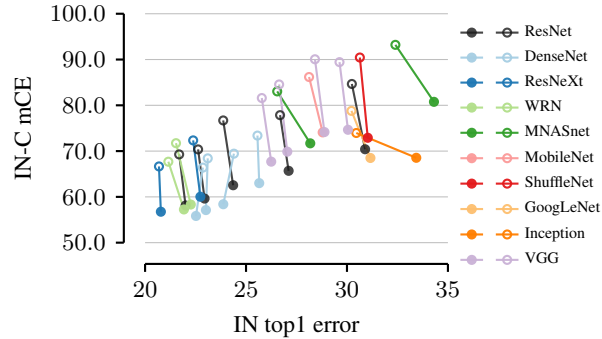

Figure 1: Sample size vs. performance tradeoff in terms of the mean corruption error (mCE) on IN-C for ResNet-50 and AugMix (AM). Black line corresponds to (non-adapted) ResNet50 state-of-the-art performance of DeepAug+AugMix.

Figure 2: Across 25 model architectures in the `torchvision` library, the baseline mCE (◦) improves with adaptation (●), often on the order of 10 points. Best viewed in color.

The hyperparameter $N$ controls the trade-off between source and estimated target statistics and has the intuitive interpretation of a *pseudo sample size* (p. 117, 14) for samples from the training set. The case $N \to \infty$ ignores the test set statistics and is equivalent to the standard ad hoc scenario while $N = 0$ ignores the training statistics. Supported by empirical and theoretical results (see results section and appendix), we suggest using $N \in [8, 128]$ for practical applications with small $n < 32$.

## 4   Experimental Setup

**Models.**    We consider a large range of models (cf. Table 2, §B,E) and evaluate pre-trained variants of DenseNet [15], GoogLeNet [16], Inception and GoogLeNet [17], MNASnet [18], MobileNet [19], ResNet [20], ResNeXt [21], ShuffleNet [22], VGG [23] and Wide Residual Network [WRN, 24] from the `torchvision` library [25]. All models are trained on the ILSVRC2012 subset of IN comprised of 1.2 million images in the training and a total of 1000 classes [7, 8]. We also consider a ResNeXt-101 variant pre-trained on a 3.5 billion image dataset and then fine-tuned on the IN training set [26]. We evaluate 3 models from the SimCLRv2 framework [27]. We additionally evaluate the four leading methods from the ImageNet-C leaderboard, namely Stylized ImageNet training [SIN; 28], adversarial noise training [ANT; 29] as well as a combination of ANT and SIN [29], optimized data augmentation using AutoAugment [AugMix; 30, 31] and Assemble Net [32]. For partial adaptation, we choose $N \in \{2^0, \cdots, 2^{10}\}$ and select the optimal value on the holdout corruption mCE.

**Datasets.**    ImageNet-C [IN-C; 2] is comprised of corrupted versions of the 50 000 images in the IN validation set. The dataset offers five severities per corruption type, for a total of 15 "test" and 4 "holdout" corruptions. ImageNet-A [IN-A; 33] consists of unmodified real-world images which yield chance level classification performance in IN trained ResNet-50 models. ImageNet-V2 [IN-V2; 34] aims to mimic the test distribution of IN, with slight differences in image selection strategies. ObjectNet [ON; 35] is a test set containing 50 000 images like IN organized in 313 object classes with 109 unambiguously overlapping IN classes. ImageNet-R [IN-R; 36] contains 30 000 images with various artistic renditions of 200 classes of the original IN dataset. Additional information on the used models and datasets can be found in §B. For IN, we resize all images to $256 \times 256$px and take the center $224 \times 224$px crop. For IN-C, images are already cropped. We also center and re-scale the color values with $\mu_{RGB} = [0.485, 0.456, 0.406]$ and $\sigma = [0.229, 0.224, 0.225]$.

## 5   Results

**Adaptation boosts robustness of a vanilla trained ResNet-50 model.**    We consider the pre-trained ResNet-50 architecture from the `torchvision` library and adapt the running mean and variance on all corruptions and severities of IN-C for different batch sizes. The results are displayed in Fig. 1 where different line styles of the green lines show the number of pseudo-samples $N$ indicating the

Table 1: Adaptation improves mCE (lower is better) and Top1 accuracy (higher is better) on IN-C for different models and surpasses the previous state of the art without adaptation. We consider $n = 8$ for partial adaptation.

| Model | IN-C mCE ($\searrow$) | | | | Top1 accuracy ($\nearrow$) | | | |
|---|---|---|---|---|---|---|---|---|
| | w/o adapt | partial adapt | full adapt | $\Delta$ | w/o adapt | partial adapt | full adapt | $\Delta$ |
| Vanilla ResNet-50 | 76.7 | 65.0 | 62.2 | $(-14.5)$ | 39.2 | 48.6 | 50.7 | $(+11.5)$ |
| SIN [28] | 69.3 | 61.5 | 59.5 | $(-9.8)$ | 45.2 | 51.6 | 53.1 | $(+7.9)$ |
| ANT [29] | 63.4 | 56.1 | 53.6 | $(-9.8)$ | 50.4 | 56.1 | 58.0 | $(+7.6)$ |
| ANT+SIN [29] | 60.7 | 55.3 | 53.6 | $(-7.0)$ | 52.6 | 56.8 | 58.0 | $(+5.4)$ |
| AugMix [AM; 30] | 65.3 | 55.4 | 51.0 | $(-14.3)$ | 48.3 | 56.3 | 59.8 | $(+11.4)$ |
| Assemble Net [32] | 52.3 | – | 50.1 | $(-1.2)$ | 59.2 | – | 60.8 | $(+1.5)$ |
| DeepAug [36] | 60.4 | 52.3 | 49.4 | $(-10.9)$ | 52.6 | 59.0 | 61.2 | $(+8.6)$ |
| DeepAug+AM [36] | 53.6 | 48.4 | 45.4 | $(-8.2)$ | 58.1 | 62.2 | 64.5 | $(+6.4)$ |
| DeepAug+AM+RNXt101 [36] | **44.5** | 40.7 | **38.0** | $(-6.6)$ | **65.2** | 68.2 | **70.3** | $(+5.1)$ |

influence of the prior given by the training statistics. With $N = 16$, we see that even adapting to a single sample can suffice to increase robustness, suggesting that even the ad hoc evaluation scenario can benefit from adaptation. If the training statistics are not used as a prior ($N = 0$), then it takes around 8 samples to surpass the performance of the non-adapted baseline model (76.7% mCE). After around 16 to 32 samples, the performance quickly converges to 62.2% mCE, considerably improving the baseline result. These results highlight the practical applicability of batch norm adaptation in basically all application scenarios, independent of the number of available test samples.

**Adaptation consistently improves corruption robustness across IN trained models.** To evaluate the interaction between architecture and BN adaptation, we evaluate all 25 pre-trained models in the `torchvision` package and visualize the results in Fig. 2. All models are evaluated with $N = 0$ and $n = 2000$. We group models into different families based on their architecture and observe consistent improvements in mCE for all of these families, typically on the order of 10% points. We observe that in both evaluation modes, DenseNets [15] exhibit higher corruption robustness despite having a comparable or even smaller number of trainable parameters than ResNets which are usually considered as the relevant baseline architecture. A take-away from this study is thus that model architecture alone plays a significant role for corruption robustness and the ResNet architecture might not be the optimal choice for practical applications.

**Adaptation yields new state of the art on IN-C for robust models.** We now investigate if BN adaptation also improves the most robust models on IN-C. The results are displayed in Table 1. All models are adapted using $n = 50\,000$ (vanilla) or $n = 4096$ (all other models) and $N = 0$. The performance of all models is considerably higher whenever the BN statistics are adapted. The DeepAugment+AugMix reaches a new state of the art on IN-C for a ResNet-50 architecture of 45.4% mCE. Evaluating the performance of AugMix over the number of samples for adaptation (Fig. 1, we find that as little as eight samples are sufficient to improve over AssembleNet [32], the current state-of-the-art ResNet-50 model on IN-C without adaptation. We have included additional results in §C.

## 6   Analysis and Ablation Studies

**Severity of covariate shift correlates with performance degradation.** The relationship between the performance degradation on IN-C and the covariate shift suggests an unsupervised way of estimating the classification performance of a model on a new corruption. Taking the normalized Wasserstein distance (cf. §A) between the statistics of the source and target domains[3] computed on all samples with the same corruption and severity and averaged across all network layers, we find a correlation with the top-1 error (Fig. 3 *i–iii*) of both non-adapted (*i*) and fully adapted model (*ii*) on IN-C corruptions. Within single corruption categories (noise, blur, weather, and digital), the relationship between top-1 error and Wasserstein distance is particularly striking: using linear

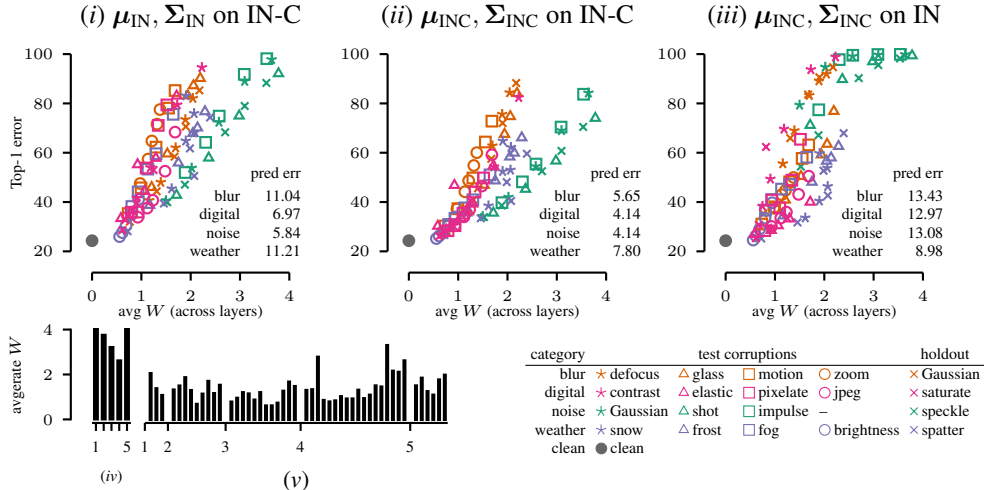

Figure 3: The Wasserstein metric between optimal source (IN) and target (IN-C) statistics correlates well with top-1 errors (*i*) of non-adapted models on IN-C, (*ii*) of adapted models on IN-C, indicating that even after reducing covariate shift, the metric is predictive of the remaining source–target mismatch (*iii*) IN-C adapted models on IN, the reverse case of (*i*). Holdout corruptions can be used to get a linear estimate on the prediction error of test corruptions (tables). We depict input and downsample (*iv*) as well as bottlneck layers (*v*) and notice the largest shift in early and late downsampling layers. The metric is either averaged across layers (*i–iii*) or across corruptions (*iv–v*).

Table 2: Improvements from adapting the BN parameters vanish for models trained with weakly supervised pre-training.

| ResNeXt101 | IN-C mCE ($\searrow$) | |
| --- | --- | --- |
| | BN | BN+adapt |
| 32x8d, IN | 66.6 | 56.7 ($-9.9$) |
| 32x8d, IG-3.5B | 51.7 | 51.6 ($-0.1$) |
| 32x48d, IG-3.5B | **45.7** | 47.3 ($+1.6$) |

Table 3: Fixup and GN trained models perform better than non-adapted BN models but worse than adapted BN models.

| Model | IN-C mCE ($\searrow$) | | | |
| --- | --- | --- | --- | --- |
| | Fixup | GN | BN | BN+adapt |
| ResNet-50 | 72.0 | 72.4 | 76.7 | **62.2** |
| ResNet-101 | 68.2 | 67.6 | 69.0 | **59.1** |
| ResNet-152 | 67.6 | 65.4 | 69.3 | **58.0** |

regression, the top-1 accuracy of hold-out corruptions can be estimated with around 1–2% absolute mean deviation (cf. §C.5) within a corruption, and with around 5–15% absolute mean deviation when the estimate is computed on the holdout corruption of each category (see Fig. 3, typically, a systematic offset remains). In Fig. 3(*iv–v*), we display the Wasserstein distance across individual layers and observe that the covariate shift is particularly present in early and late downsampling layers of the ResNet-50.

**Large scale pre-training alleviates the need for adaptation.** Computer vision models based on the ResNeXt architecture [21] pretrained on a much larger dataset comprised of $3.5 \times 10^9$ Instagram images (IG-3.5B) achieve a 45.7% mCE on IN-C [26, 37]. We re-evaluate these models with our proposed paradigm and summarize the results in Table 2. While we see improvements for the small model pre-trained on IN, these improvements vanish once the model is trained on the full IG-3.5B dataset. This observation also holds for the largest model, suggesting that training on very large datasets might alleviate the need for covariate shift adaptation.

**Group Normalization and Fixup Initialization performs better than non-adapted batch norm models, but worse than batch norm with covariate shift adaptation.** So far, we considered image classification models with BN layers and concluded that using training dataset statistics in BN generally degrades model performance in out-of-distribution evaluation settings. We now consider models trained without BN and study the impact on corruption robustness, similar to Galloway et al. [38].

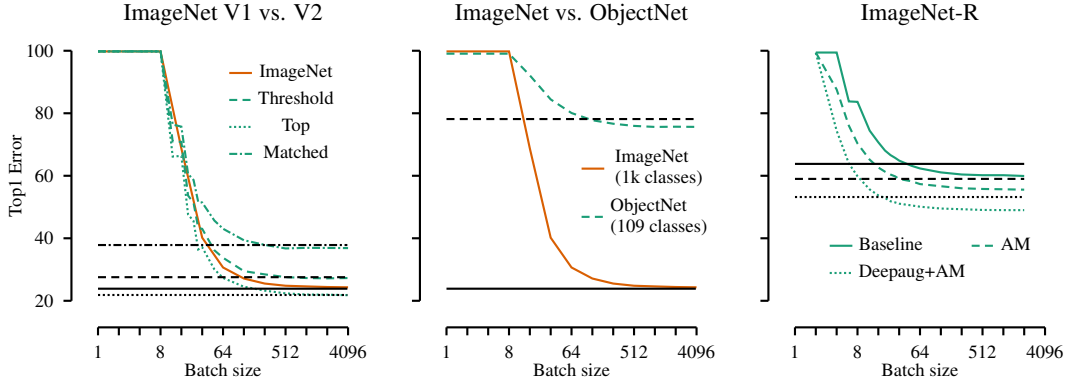

Figure 4: Batch size vs. performance trade-off for different natural image datasets with no covariate shift (IN, IN-V2), complex and shuffled covariate shift (ObjectNet), complex and systematic covariate shift (ImageNet-R). Straight black lines show baseline performance (no adaptation). ImageNet plotted for reference.

Table 4: GN and Fixup achieve the best results on ObjectNet (ON). After shuffling IN-C corruptions, BN adaptation does no longer decrease the error. Adaptation improves the performance of a vanilla ResNet50 on IN-R.

| ResNet50 | ON | | Mixed IN-C | | IN-R |
|---|---|---|---|---|---|
| | top-1 | top-5 | top-1 | top-5 | top-1 |
| BN w/o adapt | 78.2 | 60.9 | 61.1 | 40.8 | 63.8 |
| BN w/ adapt | 76.0 | 58.9 | 60.9 | 40.3 | **59.9** |
| GroupNorm | **70.8** | **49.8** | 57.3 | 36.0 | 61.2 |
| Fixup | 71.5 | 51.4 | **56.8** | **35.4** | 65.0 |

Table 5: Adaptation improves the performance (top-1 error) of robust models on IN-R (n=2048).

| Model | base | adapt | $\Delta$ |
|---|---|---|---|
| ResNet50 | 63.8 | 59.9 | -3.9 |
| SIN | 58.6 | 54.2 | -4.4 |
| ANT | 61.0 | 58.0 | -3.0 |
| ANT+SIN | 53.8 | 52.0 | -1.8 |
| AugMix (AM) | 59.0 | 55.8 | -3.2 |
| DeepAug (DAug) | 57.8 | 52.5 | -5.3 |
| DAug+AM | 53.2 | 48.9 | -4.3 |
| DAug+AM+RNXt101 | **47.9** | **44.0** | -3.9 |

First, using Fixup initialization [39] alleviates the need for BN layers. We train a ResNet-50 model on IN for 100 epochs to obtain a top-1 error of 24.2% and top-5 error of 7.6% (compared to 27.6% reported by Zhang et al. [39] with shorter training, and the 23.9% obtained by our ResNet-50 baseline trained with BN). The model obtains an IN-C mCE of 72.0% compared to 76.7% mCE of the vanilla ResNet-50 model and 62.2% mCE of our adapted ResNet-50 model (cf. Table 3). Additionally, we train a ResNet-101 and a ResNet-152 with Fixup initialization with similar results. Second, GroupNorm [GN; 40] has been proposed as a batch-size independent normalization technique. We train a ResNet-50, a ResNet-101 and a ResNet-152 architecture for 100 epochs and evaluate them on IN-C and find results very similar to Fixup.

**Results on other datasets: IN-A, IN-V2, ObjectNet, IN-R** We use $N = 0$ and vary $n$ in all ablation studies in this subsection. The technique does not work for the case of "natural adversarial examples" of IN-A [33] and the error rate stays above 99%, suggesting that the covariate shift introduced in IN-A by design is more severe compared to the covariate shift of IN-C and can not be corrected by merely calculating the correct BN statistics. We are not able to increase performance neither on IN nor on IN-V2, since in these datasets, no domain shift is present by design (see Fig. 4). For ON, the performance increases slightly when computing statistics on more than 64 samples. In Table 4 (first and second column), we observe that the GroupNorm and Fixup models perform better than our BN adaptation scheme: while there is a dataset shift in ON compared to IN, BN adaptation is only helpful for *systematic* shifts across multiple inputs and this assumption is violated on ON. As a control experiment, we sample a dataset "Mixed IN-C" where we shuffle the corruptions and severities. In Table 4 (third and fourth column), we now observe that BN adaptation expectedly no longer improves performance. On IN-R, we achieve better results for the adapted model compared to the non-adapted model as well as the GroupNorm and Fixup models, see Table 4 (last column). Additionally, on IN-R, we decrease the top-1 error for a wide range of models through adaptation (see

Table 5). For IN-R, we observe performance improvements for the vanilla trained ResNet50 when using a sample size of larger than 32 samples for calculating the statistics (Fig. 4, right-most plot).

**A model for correcting covariate shift effects.** We evaluate how the batch size for estimating the statistics at test time affects the performance on IN, IN-V2, ON and IN-R in Fig. 4. As expected, for IN the adaptation to test time statistics converges to the performance of the train time statistics in the limit of large batch sizes, see Fig. 4 middle. For IN-V2, we find similar results, see Fig. 4 left. This observation shows that (*i*) there is no systematic covariate shift between the IN train set and the IN-V2 validation set that could be corrected by using the correct statistics and (*ii*) is further evidence for the *i.i.d.* setting pursued by the authors of IN-V2. In case of ON (Fig. 4 right), we see slight improvements when using a batch size bigger than 128.

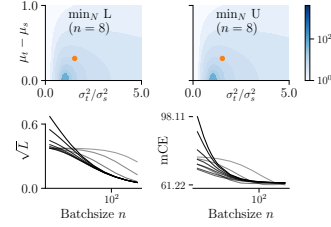

Figure 5: The bound suggests small optimal $N$ for most parameters (i) and qualitatively explains our empirical observation (ii).

Choosing the number of pseudo-samples $N$ offers an intuitive trade-off between estimating accurate target statistics (low $N$) and relying on the source statistics (large $N$). We propose a simple model to investigate optimal choices for $N$, disregarding all special structure of DNNs, and focusing on the statistical error introduced by estimating $\hat{\mu}_t$ and $\hat{\sigma}_t^2$ from a limited number of samples $n$. To this end, we estimate upper ($U$) and lower ($L$) bounds of the expected squared Wasserstein distance $W_2^2$ as a function of $N$ and the covariate shift which provides good empirical fits between the estimated $W$ and empirical performance for ResNet-50 for different $N$ (Fig. 5; bottom row). Choosing $N$ such that $L$ or $U$ are minimized (Fig. 5; example in top row) qualitatively matches the values we find, see §D for all details.

**Proposition 1** (Bounds on the expected value of the Wasserstein distance between target and combined estimated target and source statistics)**.** *We denote the source statistics as $\mu_s, \sigma_s^2$, the true target statistics as $\mu_t, \sigma_t^2$ and the biased estimates of the target statistics as $\hat{\mu}_t, \hat{\sigma}_t^2$. For normalization, we take a convex combination of the source statistics and estimated target statistics as discussed in Eq. 3. At a confidence level $1 - \alpha$, the expectation value of the Wasserstein distance $W_2^2(\bar{\mu}, \bar{\sigma}, \mu_t, \sigma_t)$ between ideal and estimated target statistics w.r.t. to the distribution of sample mean $\hat{\mu}_t$ and sample variance $\hat{\sigma}_t^2$ is bounded from above and below with $L \leq \mathbb{E}[W_2^2] \leq U$, where*

$$L = \left( \sigma_t - \sqrt{\frac{N}{N+n}\sigma_s^2 + \frac{n-1}{N+n}\sigma_t^2} \right)^2 + \frac{N^2}{(N+n)^2}(\mu_t - \mu_s)^2 + \frac{n}{(N+n)^2}\sigma_t^2$$

$$U = L + \sigma_t^5 \frac{(n-1)}{2(N+n)^2} \left( \frac{N}{N+n}\sigma_s^2 + \frac{1}{N+n}\chi^2_{1-\alpha/2,n-1}\sigma_t^2 \right)^{-3/2}$$

*The quantity $\chi^2_{1-\alpha/2,n-1}$ denotes the left tail value of a chi square distribution with $n-1$ degrees of freedom, defined as $P\left( X \leq \chi^2_{1-\alpha/2,n-1} \right) = \alpha/2$ for $X \sim \chi^2_{n-1}$.* Proof: *See Appendix §D.*

## 7 Related Work

The IN-C benchmark [2] has been extended to MNIST [41], several object detection datasets [42] and image segmentation [43] reflecting the interest of the robustness community. Most proposals for improving robustness involve special training protocols, requiring time and additional resources. This includes data augmentation like Gaussian noise [44], optimized mixtures of data augmentations in conjunction with a consistency loss [30], training on stylized images [28, 42, 45] or against adversarial noise distributions [29]. Other approaches tweak the architecture, e.g. by adding shift-equivariance with an anti-aliasing module, [46] or assemble different training techniques [32].

Unsupervised domain adaptation (DA) is a form of transductive inference where additional information about the test dataset is used to adapt a model to the test distribution. Adapting feature statistics was proposed by Sun et al. [47] and follow up work evaluated the performance of adapting BN parameters in unsupervised [5, 6] and supervised DA settings [4]. As an application example in medical imaging, Bug et al. [48] show that adaptive normalization is useful for removing domain

shifts on histopathological data. More involved methods for DA include self-supervised domain adaptation on single examples [49] and pseudo-labeling French et al. [50]. Xie et al. [51] achieve the state of the art on IN-C with pseudo-labeling. In work concurrent to ours, Wang et al. [52] also show BN adaptation results on IN-C. They also perform experiments on CIFAR10-C and CIFAR100-C and explore other domain adaptation techniques.

Robustness scores obtained by adversarial training can be improved when separate BN or GroupNorm layers are used for clean and adversarial images [53]. The expressive power of adapting only affine BN parameters BN parameters was shown in multi-task [54] and DA contexts [4] and holds even for fine-tuning randomly initialized ResNets [55]. Concurrent work shows additional evidence that BN adaptation yields increased performance on ImageNet-C [56].

## 8  Discussion and Conclusion

We showed that reducing covariate shift induced by common image corruptions improves the robustness of computer vision models trained with BN layers, typically by 10–15% points (mCE) on IN-C. Current state-of-the-art models on IN-C can benefit from adaptation, sometimes drastically like AugMix ($-14\%$ points mCE). This observation underlines that current benchmark results on IN-C underestimate the corruption robustness that can be reached in many application scenarios where additional (unlabeled) samples are available for adaptation.

Robustness against common corruptions improves even if models are adapted only to a single sample, suggesting that BN adaptation should always be used whenever we expect machine vision algorithms to encounter out-of-domain samples. Most further improvements can be reaped by adapting to 32 to 64 samples, after which additional improvements are minor.

Our empirical results suggest that the performance degradation on corrupted images can mostly be explained by the difference in feature-wise first and second order moments. While this might sound trivial, the performance could also degrade because models mostly extract features susceptible to common corruptions [57], which could not be fixed without substantially adapting the model weights. The fact that model robustness increases after correcting the BN statistics suggests that the features upon which the models rely on are still present in the corrupted images. The opposite is true in other out-of-domain datasets like IN-A or ObjectNet where our simple adaptation scheme does not substantially improve performance, suggesting that here the main problem is in the features that models have learned to use for prediction.

Batch Norm itself is not the reason why models are susceptible to common corruptions. While alternatives like Group Normalization and Fixup initialization slightly increase robustness, the adapted BN models are still substantially more robust. This suggests that non-BN models still experience an internal covariate shift on corrupted images, but one that is now absorbed by the model parameters instead of being exposed in the BN layers, making it harder to fix.

Large-scale pre-training on orders of magnitude more data (like IG-3.5B) can remove the first- and second-order covariate shift between clean and corrupted image samples, at least partially explaining why models trained with weakly supervised training [26] generalize so well to IN-C.

Current corruption benchmarks emphasize ad hoc scenarios and thus focus and bias future research efforts on these constraints. Unfortunately, the ad hoc scenario does not accurately reflect the information available in many machine vision applications like classifiers in medical computer vision or visual quality inspection algorithms, which typically encounter a similar corruption continuously and could benefit from adaptation. This work is meant to spark more research in this direction by suggesting two suitable evaluation metrics—which we strongly suggest to include in all future evaluations on IN-C—as well as by highlighting the potential that even a fairly simple adaptation mechanism can have for increasing model robustness. We envision future work to also adopt and evaluate more powerful domain adaptation methods on IN-C and to develop new adaptation methods specifically designed to increase robustness against common corruptions.

## Broader Impact

The primary goal of this paper is to increase the robustness of machine vision models against common corruptions and to spur further progress in this area. Increasing the robustness of machine vision

systems can enhance their reliability and safety, which can potentially contribute to a large range of use cases including autonomous driving, manufacturing automation, surveillance systems, health care and others. Each of these uses may have a broad range of societal implications: autonomous driving can increase mobility of the elderly and enhance safety, but could also enable more autonomous weapon systems. Manufacturing automation can increase resource efficiency and reduce costs for goods, but may also increase societal tension through job losses or increase consumption and thus waste. Of particular concern (besides surveillance) is the use of generative vision models for spreading misinformation or for creating an information environment of uncertainty and mistrust.

We encourage further work to understand the limitations of machine vision models in out-of-distribution generalization settings. More robust models carry the potential risk of automation bias, i.e., an undue trust in vision models. However, even if models are robust to common corruptions, they might still quickly fail on slightly different perturbations like surface reflections. Understanding under what conditions model decisions can be deemed reliable or not is still an open research question that deserves further attention.

## Acknowledgments and Disclosure of Funding

We thank Julian Bitterwolf, Roland S. Zimmermann, Lukas Schott, Mackenzie W. Mathis, Alexander Mathis, Asim Iqbal, David Klindt, Robert Geirhos, other members of the Bethge and Mathis labs and four anonymous reviewers for helpful suggestions for improving our manuscript and providing ideas for additional ablation studies. We thank the International Max Planck Research School for Intelligent Systems (IMPRS-IS) for supporting E.R. and St.S.; St.S. acknowledges his membership in the European Laboratory for Learning and Intelligent Systems (ELLIS) PhD program. This work was supported by the German Federal Ministry of Education and Research (BMBF) through the Tübingen AI Center (FKZ: 01IS18039A), by the Deutsche Forschungsgemeinschaft (DFG) in the priority program 1835 under grant BR2321/5-2 and by SFB 1233, Robust Vision: Inference Principles and Neural Mechanisms (TP3), project number: 276693517. The authors declare no conflicts of interests.

## Footnotes

\*Equal contribution. [†] Equal contribution.; Online version and code: domainadaptation.org/batchnorm

[2]Note that our notion of internal covariate shift differs from previous work [3, 11]: In *i.i.d.* training settings, Ioffe and Szegedy [3] hypothesized that covariate shift introduced by changing lower layers in the network is reduced by BN, explaining the empirical success of the method. We do not provide evidence for this line of research in this work: Instead, we focus on the covariate shift introduced (by design) in datasets such as IN-C, and provide evidence for the hypothesis that high-level domain shifts in the input partly manifests in shifts and scaling of *internal* activations.

[3]For computing the Wasserstein metric we make the simplifying assumption that the empirical mean and covariances fully parametrize the respective distributions.

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
