[Supplementary Material]

# Supplementary Material

# A Distances and divergences for quantifying domain shift

Besides analyzing the performance drop when evaluating a model using source statistics on a target dataset, we consider the mismatch in model statistics directly. We first take an ImageNet trained model and adapt it to each of the 95 conditions in IN-C. To obtain a more exact estimate of the true statistics, we split the model into multiple stages with only few BN layers per stage and apply the following simple algorithm[4]:

- Start with image inputs $\mathbf{z}_n^0 \leftarrow \mathbf{x}_n$ from the validation set to adapt to, for each $n \in [50000]$.
- Split the model into multiple stages, $h(\mathbf{x}) = (f_m \circ \cdots \circ f_1)(\mathbf{x})$, where each module $f_i$ can potentially contain one or multiple BN layers. We denote the number of BN layers in the $i$-th module as $b_i$.
- For each stage $i \in [m]$, repeat $b_i$ times: $\mathbf{z}_n^i \leftarrow f_i(\mathbf{z}_n^{i-1})$ for each $n$, and update the BN statistics in module $f_i(\mathbf{z}_n^{i-1})$.
- Return $h$ with adapted statistics.

Using this scheme, we get source statistics $\mu_s$ and $\Sigma_s$ for each layer and $\mu_t$ and $\Sigma_t$ for each layer and corruption. In total, we get 96 different collections of statistics across network layers (for IN and the 95 conditions in IN-C). For simplicity, we will not further index the statistics. Note that all covariance matrices considered here are diagonal, which is a further simplification. We expect that our domain shift estimates could be improved by considering the full covariance matrices.

In the following, we will introduce three possible distances and divergences which can be applied between source and target statistics to quantify the effect of common corruptions induced covariate shift. We consider the Wasserstein distance, a normalized version of the Wasserstein distance, and the Jeffrey divergence.

## A.1 The Wasserstein distance

Given a baseline ResNet-50 model with source statistics $\mu_s, \Sigma_s$ on IN, the Wasserstein distance (cf. 58) between the train and test distribution with statistics $\mu_t, \Sigma_t$ is given as

$$W_2(p_s, p_t)^2 = \|\boldsymbol{\mu}_s - \boldsymbol{\mu}_t\|_2^2 + \mathrm{tr}\left(\boldsymbol{\Sigma}_s + \boldsymbol{\Sigma}_t - 2\left(\boldsymbol{\Sigma}_t^{1/2}\boldsymbol{\Sigma}_s\boldsymbol{\Sigma}_t^{1/2}\right)^{1/2}\right). \tag{4}$$

## A.2 The source-normalized Wasserstein distance

When estimated for multiple layers across the network, the Wasserstein distance between source and target depends on the overall magnitude of the statistics. Practically, this means the metric is dominated by features with large magnitude (e.g. in the first layer of a neural network, which receives larger inputs).

To mitigate this issue, we normalize both statistics with the source statistics and define the normalized Wasserstein distance as

$$\widetilde{W}_2^2 = W_2^2\left(\boldsymbol{\Sigma}_s^{-1/2}\boldsymbol{\mu}_s, \mathbf{I}, \boldsymbol{\Sigma}_s^{-1/2}\boldsymbol{\mu}_t, \boldsymbol{\Sigma}_s^{-1}\boldsymbol{\Sigma}_t\right) \tag{5}$$

$$= \mathrm{Tr}\left(\mathbf{I} + \boldsymbol{\Sigma}_t\boldsymbol{\Sigma}_s^{-1} - 2\boldsymbol{\Sigma}_t^{1/2}\boldsymbol{\Sigma}_s^{-1/2}\right) + (\boldsymbol{\mu}_t - \boldsymbol{\mu}_s)^T\boldsymbol{\Sigma}_s^{-1}(\boldsymbol{\mu}_t - \boldsymbol{\mu}_s). \tag{6}$$

In the uni-variate case, the normalized Wasserstein distance $\widetilde{W}_2^2$ is equal to the Wasserstein distance $W_2^2$ between source and target statistics divided by $\sigma_s^2$:

$$\widetilde{W}_2^2 = W_2^2\left(\frac{\mu_s}{\sigma_s}, 1, \frac{\mu_t}{\sigma_s}, \frac{\sigma_t^2}{\sigma_s^2}\right) = 1 + \frac{\sigma_t^2}{\sigma_s^2} - 2\frac{\sigma_t}{\sigma_s} + \frac{(\mu_t - \mu_s)^2}{\sigma_s^2} = \frac{1}{\sigma_s^2}W_2^2(\mu_s, \sigma_s^2, \mu_t, \sigma_t^2). \tag{7}$$

## A.3 The Jeffrey divergence

The Jeffrey divergence $J(p_s, p_t)$ between source distribution $p_s$ and target distribution $p_t$ is the symmetrized version of the Kullback-Leibler divergence $D_{KL}$:

$$J(p_s, p_t) = \frac{1}{2} \left( D_{KL}(p_s \| p_t) + D_{KL}(p_t \| p_s) \right) \tag{8}$$

The Kullback-Leibler divergence between the $D$-dimensional multivariate normal source and target distributions is defined as

$$D_{KL}(\mathcal{N}_t \| \mathcal{N}_s) = \frac{1}{2} \left( \mathrm{Tr}\left( \mathbf{\Sigma}_s^{-1} \mathbf{\Sigma}_t \right) + (\boldsymbol{\mu}_s - \boldsymbol{\mu}_t)^\top \mathbf{\Sigma}_s^{-1} (\boldsymbol{\mu}_s - \boldsymbol{\mu}_t) - D + \ln\left( \frac{\det \mathbf{\Sigma}_s}{\det \mathbf{\Sigma}_t} \right) \right). \tag{9}$$

The Jeffrey divergence between the $D$-dimensional multivariate normal source and target distributions then follows as

$$J(\mathcal{N}_t, \mathcal{N}_s) = \frac{1}{4} \left( \mathrm{Tr}\left( \mathbf{\Sigma}_s^{-1} \mathbf{\Sigma}_t \right) + \mathrm{Tr}\left( \mathbf{\Sigma}_t^{-1} \mathbf{\Sigma}_s \right) + (\boldsymbol{\mu}_s - \boldsymbol{\mu}_t)^\top \left( \mathbf{\Sigma}_s^{-1} + \mathbf{\Sigma}_t^{-1} \right) (\boldsymbol{\mu}_s - \boldsymbol{\mu}_t) - 2D \right). \tag{10}$$

## A.4 Summary statistics and quantification of covariate shift between different IN-C conditions

Given the 95 distances/divergences between the baseline (IN) statistics and 95 IN-C conditions, we first perform a layer-wise analysis of the statistics and depict the results in Figure 6. The unnormalized Wasserstein distance is sensitive to the magnitude of the source statistics and hence differs qualitatively from the results on the normalized Wasserstein distance and Jeffrey Divergence. We appreciate that the most notable difference between source and target domains is visible in the ResNet-50 downsampling layers. All three metrics suggest that the shift is mainly present in the first and final layers of the network, supporting the hypothesis that within the common corruption dataset, we have both superficial covariate shift which can be corrected by simple means (such as brightness or contrast variations) in the first layers, and also more "high-level" domain shifts which can only be corrected in the later layers of the network.

In Figure 7, we more closely analyze this relationship for different common corruptions. We can generally appreciate the increased measures as the corruption severity increases.

Figure 6: Wasserstein distance, normalized Wasserstein distance and Jeffrey divergence estimated among source and target statistics between different network layers. We report the respective metric w.r.t. to the difference between baseline (IN) and target (IN-C) statistics and show the value averaged across all corruptions. We note that for a ResNet-50 model, downsampling layers contribute most to the overall error.

Figure 7: Normalized Wasserstein distance and Jeffrey divergence across corruptions and layers in a ResNet-50.

# B Notes on the experimental setup

## B.1 Practical considerations for implementing the method

Our method is conceptually very easy to implement. We generally recommend to first explore the easier variant of the algorithm where $N = 0$, i.e., no source statistics are used. As shown in our experiments, this setting works well if 100 or more target samples are available.

In this case, implementing the method boils down to enabling the training mode for all BN layers across the network. We will discuss this option along with two variants important for application to practical problems: Using exponential moving averaging (EMA) to collect target statistics across multiple batches, and using the source statistics as a prior.

**Example implementation in PyTorch and caveats**   We encourage authors of robust models to always evaluate their models, and in particular baseline algorithms on both the train and test set statistics. Implementation in both PyTorch, Tensorflow and other machine learning libraries is straightforward and adds only minimal overhead. For PyTorch, adaptation is possible by simply adding

```
def use_test_statistics(module):
  if isisinstance(module, nn._BatchNorm):
    module.train()
model.eval()
model.apply(use_test_statistics)
```

before starting a model evaluation. For the adaptation to a full dataset, we provide a reference implementation with the source code release of this paper. Also, in contrast to the convention of not shuffling examples during test time, *make sure to enable dataset shuffling also during test time* in order to compute the correct statistics marginalized over class assignment.

**Exponential moving averaging**   In practice, it might be beneficial to keep track of samples already encountered and use a running mean and variance on the test set to normalize new samples. We can confirm that this technique closely matches the full-dataset adaptation case even when evaluating with batch size 1 and is well suited for settings with less powerful hardware, or in general settings where access to the full batch of samples is not possible. Variants of this technique include the adaptation of the decay factor to discard statistics of samples encountered in the past (e.g. when the data domain slowly drifts over time).

## B.2 Notes on models

Note that we only re-evaluate existing model checkpoints, and hence do not perform any hyperparameter tuning or adaptations to model training except for selecting the pseudo batchsize $N$ for the source domain. Depending on the batch size and the architecture, model evaluations are done on one to eight Nvidia RTX 2080 GPUs (i.e., using 12 to 96 GB of memory) or up to four Nvidia V100 GPUs (128 GB of memory). Since we merely re-evaluate trained models, it is also possible to work on less powerful hardware with less memory. In these cases, the aggregation of batch normalization statistics has to be done across several batches using a variant of EMA.

## B.3 Hyperparameter tuning

Our method is generally parameter-free if only target statistics should be considered for normalization. This approach is generally preferred for larger batch sizes $n$ and should also be adapted in practice when a sufficient amount of samples is available. For tuning $N$, we consider the pre-defined holdout corruptions in IN-C, including speckle noise, saturation, Gaussian blur and spatter using a grid search across different values for $N$.

## B.4 Notes on datasets

In the main paper, we have used several datasets and provide more relevant information here:

**ImageNet-C (IN-C)** For the evaluation on IN-C, we use the JPEG compressed images from github.com/hendrycks/robustness as is advised by the authors to ensure reproducibility. We note that Ford et al. [44] report a decrease in performance when the compressed JPEG files are used as opposed to applying the corruptions directly in memory without compression artefacts.

**ObjectNet (ON)** We find that there are 9 classes with multiple possible mappings from ON to IN (see the list in Table 6); we discard these classes in our evaluation. Models trained on IN experience a large performance drop on the order of 40–45% when tested on ON. ON is an interesting test case for unsupervised domain adaptation since IN and ON are likely sampled from different distributions. ON intentionally shows objects from new viewpoints on new backgrounds.

**ImageNet-V2 (IN-V2)** There are three test sets in IN-V2 that differ in *selection frequencies* of the MTurk workers. The selection frequency is given by the fraction of MTurk workers who selected an image for its target class. For the "MatchedFrequency" dataset, images were sampled according to the estimated selection frequency of sampling of the original IN validation dataset. For the "Threshold0.7" variant of IN-V2, images were sampled with a selection frequency of at least 0.7. The "TopImages" was sampled from images with the highest selection frequency. Although all three test sets were sampled from the same Flickr candidate pool and were labeled correctly and selected by more than 70% of MTurk workers, the model accuracies on these datasets vary by 14%. The authors observe a systematic accuracy drop when comparing model performance on the original IN validation set and IN-V2 and attribute it to the distribution gap between their datasets and the original IN dataset. They quantify the distribution gap by how much the change from the original distribution to the new distribution affects the considered model. Engstrom et al. analyze the creation process of IN-V2 and identify statistical bias resulting from noisy readings of the selection frequency statistic as a main source of dropping performance [59]. After correcting the bias, [59] find that the accuracy drop between IN and IN-V2 measures only 3.6% ± 1.5% of the original 11.7% ± 1.0%.

| ON class | IN classes |
|---|---|
| wheel | wheel; paddlewheel, paddle wheel |
| helmet | football helmet; crash helmet |
| chair | barber chair; folding chair; rocking chair, rocker |
| still_camera | Polaroid camera, Polaroid Land camera; reflex camera |
| alarm_clock | analog clock; digital clock |
| tie | bow tie, bow-tie, bowtie; Windsor tie |
| pen | ballpoint, ballpoint pen, ballpen, Biro; quill, quill pen; fountain pen |
| bicycle | mountain bike, all-terrain bike, off-roader; bicycle-built-for-two, tandem bicycle, tandem |
| skirt | hoopskirt, crinoline; miniskirt, mini; overskirt |

Table 6: Mapping between 9 ambiguous ON classes and the possible correspondences in IN. Different IN classes are separated with a semicolon.

## B.5 Overview of models in torchvision

In Table 7, we provide a list of the models we evaluate in the main paper, along with numbers of trainable parameters and BN parameters. Note that the fraction of BN parameters is at most at 1% compared to all trainable parameters in all considered models.

## B.6 Baseline corruption errors

In Table 8, we report the scores used for converting top-1 error into the mean corruption error (mCE) metric proposed by Hendrycks and Dietterich [2].

## B.7 Software stack

We use various open source software packages for our experiments, most notably Docker [60], scipy and numpy [61], GNU parallel [62], Tensorflow [63], PyTorch [10] and torchvision [25].

| Model | Parameter Count | BN Parameters | Fraction (%) |
|---|---|---|---|
| densenet121 | $7.98 \times 10^6$ | $8.36 \times 10^4$ | 0.010 |
| densenet161 | $2.87 \times 10^7$ | $2.20 \times 10^5$ | 0.008 |
| densenet169 | $1.41 \times 10^7$ | $1.58 \times 10^5$ | 0.011 |
| densenet201 | $2.00 \times 10^7$ | $2.29 \times 10^5$ | 0.011 |
| googlenet | $1.30 \times 10^7$ | $1.51 \times 10^4$ | 0.001 |
| inception-v3 | $2.72 \times 10^7$ | $3.62 \times 10^4$ | 0.001 |
| mnasnet0-5 | $2.22 \times 10^6$ | $2.06 \times 10^4$ | 0.009 |
| mnasnet0-75 | $3.17 \times 10^6$ | $2.98 \times 10^4$ | 0.009 |
| mnasnet1-0 | $4.38 \times 10^6$ | $3.79 \times 10^4$ | 0.009 |
| mnasnet1-3 | $6.28 \times 10^6$ | $4.88 \times 10^4$ | 0.008 |
| mobilenet-v2 | $3.50 \times 10^6$ | $3.41 \times 10^4$ | 0.010 |
| resnet101 | $4.45 \times 10^7$ | $1.05 \times 10^5$ | 0.002 |
| resnet152 | $6.02 \times 10^7$ | $1.51 \times 10^5$ | 0.003 |
| resnet18 | $1.17 \times 10^7$ | $9.60 \times 10^3$ | 0.001 |
| resnet34 | $2.18 \times 10^7$ | $1.70 \times 10^4$ | 0.001 |
| resnet50 | $2.56 \times 10^7$ | $5.31 \times 10^4$ | 0.002 |
| resnext101-32x8d | $8.88 \times 10^7$ | $2.03 \times 10^5$ | 0.002 |
| shufflenet-v2-x0-5 | $1.37 \times 10^6$ | $7.95 \times 10^3$ | 0.006 |
| shufflenet-v2-x1-0 | $2.28 \times 10^6$ | $1.62 \times 10^4$ | 0.007 |
| shufflenet-v2-x1-5 | $3.50 \times 10^6$ | $2.34 \times 10^4$ | 0.007 |
| shufflenet-v2-x2-0 | $7.39 \times 10^6$ | $3.37 \times 10^4$ | 0.005 |
| vgg11-bn | $1.33 \times 10^8$ | $5.50 \times 10^3$ | $4.142 \times 10^{-5}$ |
| vgg13-bn | $1.33 \times 10^8$ | $5.89 \times 10^3$ | $4.425 \times 10^{-5}$ |
| vgg16-bn | $1.38 \times 10^8$ | $8.45 \times 10^3$ | $6.106 \times 10^{-5}$ |
| vgg19-bn | $1.44 \times 10^8$ | $1.10 \times 10^4$ | $7.662 \times 10^{-5}$ |
| wide-resnet101-2 | $1.27 \times 10^8$ | $1.38 \times 10^5$ | 0.001 |
| wide-resnet50-2 | $6.89 \times 10^7$ | $6.82 \times 10^4$ | 0.001 |

Table 7: Overview of different models with parameter counts. We show the total number of BN parameters, which is a sum of affine parameters.

| Category | Corruption | top1 error |
|---|---|---|
| Noise | Gaussian Noise | 0.886428 |
| | Shot Noise | 0.894468 |
| | Impulse Noise | 0.922640 |
| Blur | Defocus Blur | 0.819880 |
| | Glass Blur | 0.826268 |
| | Motion Blur | 0.785948 |
| | Zoom Blur | 0.798360 |
| Weather | Snow | 0.866816 |
| | Frost | 0.826572 |
| | Fog | 0.819324 |
| | Brightness | 0.564592 |
| | Contrast | 0.853204 |
| Digital | Elastic Transform | 0.646056 |
| | Pixelate | 0.717840 |
| | JPEG Compression | 0.606500 |
| Hold-out Noise | Speckle Noise | 0.845388 |
| Hold-out Digital | Saturate | 0.658248 |
| Hold-out Blur | Gaussian Blur | 0.787108 |
| Hold-out Weather | Spatter | 0.717512 |

Table 8: AlexNet top1 errors on ImageNet-C

Table 9: After converting the checkpoints from TensorFlow to Pytorch, we notice a slight degradation in performance on the IN val set.

IN val top-1 accuracy in %.

| Model | TF | PyTorch |
|---|---|---|
| SimCLRv2 ResNet50 | 76.3 | 75.6 |
| SimCLRv2 ResNet101 | 78.2 | 77.5 |
| SimCLRv2 ResNet152 | 79.3 | 78.6 |

Table 10: Adaptation improves the performance of the ResNet50 and the ResNet101 model but hurts the performance of the ResNet152 model.

ImageNet-C (n=4096), mCE.

| Model, adaptation: | base | adapt | $\Delta$ |
|---|---|---|---|
| SimCLRv2 ResNet50 | 72.4 | 68.0 | -4.2 |
| SimCLRv2 ResNet101 | 66.6 | 65.1 | -0.9 |
| SimCLRv2 ResNet152 | 63.7 | 64.2 | +0.5 |

Figure 8: Adaptation (●) improves baseline (○) mCE across all 25 model architectures in the `torchvision` library, often on the order of 10% points. Best viewed in color.

## C  Additional results

### C.1  Performance of SimCLRv2 models

We evaluate the performance of 3 models from the SimCLRv2 framework with and without batchnorm adaptation. We test a ResNet50, a ResNet101 and a ResNet152, finetuned on 100% of IN training data. Since our code-base is in PyTorch, we use the Pytorch-SimCLR-Converter [64] to convert the provided checkpoints from Tensorflow to PyTorch. We notice a slight decline in performance when comparing the top-1 accuracy on the IN validation set, see Table 9. For preprocessing, we disable the usual PyTorch normalization and use the PIL.Image.BICUBIC interpolation for resizing because this interpolation is used in the TensorFlow code (instead of the default PIL.Image.BILINEAR in PyTorch).

The BN adaptation results for the converted models are shown in Table 10. Adaptation improves the performance of the ResNet50 and the ResNet101 model, but hurts the performance of the ResNet152 model.

### C.2  Relationship between parameter count and IN-C improvements

In addition to Fig. 3 in the main paper, we show the relationship between parameter count and IN-C mCE. In general, we see that the parameter counts correlates with corruption robustness since larger models have smaller mCE values.

### C.3  Per-corruption results on IN-C

We provide more detailed results on the individual corruptions of IN-C for the most important models considered in our study in Fig. 9. The results are shown for models where the BN parameters are

adapted on the full test sets. The adaptation consistently improves the error rates on all corruptions for both vanilla and AugMix.

Figure 9: Results on the individual corruptions of IN-C for the vanilla trained ResNet-50 and the AugMix model with and without adaptation. Adaptation reduces the error on all corruptions.

## C.4 Qualitative analysis of similarities between common corruptions

In this analysis, we compute a t-SNE embedding of the Wasserstein distances between the adapted models and the non-adapted model from Section 5, Fig. 4(i) of the main paper. The results are displayed in Fig. 10. We observe that the different corruption categories indicated by the different colors are grouped together except for the 'digital' category (pink). This visualization shows that corruption categories mostly induce similar shifts in the BN parameters. This might be an explanation why training a model on Gaussian noise generalizes so well to other noise types as has been observed by Rusak et al. [29]: By training on Gaussian noise, the BN statistics are adapted to the Gaussian noise corruption and from Fig. 10, we observe that these statistics are similar to the BN statistics of other noises.

Figure 10: t-SNE embeddings of the Wasserstein distances between BN statistics adapted on the different corruptions. This plot shows evidence on the similarities between different corruption types.

## C.5 Error prediction based on the Wasserstein distance

In Section 5, Fig. 4(i), we observe that the relationship between the Wasserstein distance and the top-1 error on IN-C is strikingly linear in the considered range of the Wasserstein distance. Similar corruptions and corruption types (indicated by color) exhibit similar slope, allowing to approximate the expected top-1 error rate without any information about the test domain itself. Using the split of the 19 corruptions into 15 test and 4 holdout corruptions [2], we compute a linear regression model on the five data points we get for each of the holdout corruptions (corresponding to the five severity levels), and use this model to predict the expected top-1 error rates for the remaining corruptions within the corruption family. This scheme works particularly for the "well defined" corruption types such as noise and digital (4.1% points absolute mean deviation from the real error. The full results are depicted in Table 11.

| | test error | | | holdout (train) error | | | model | |
|---|---|---|---|---|---|---|---|---|
| | true | pred | $|\Delta|$ | true | pred | $|\Delta|$ | coef | intercept |
| Fig. 3 (i) | | | | | | | | |
| blur | 64.89 | 54.53 | 11.04 | 58.13 | 58.13 | 3.24 | 37.59 | -0.70 |
| digital | 54.37 | 51.96 | 6.97 | 38.08 | 38.08 | 0.60 | 37.20 | 6.39 |
| noise | 73.29 | 69.68 | 5.84 | 64.51 | 64.51 | 0.65 | 24.66 | 1.68 |
| weather | 53.87 | 42.92 | 11.21 | 50.84 | 50.84 | 5.48 | 25.80 | 6.33 |
| Fig. 3 (ii) | | | | | | | | |
| blur | 55.68 | 53.28 | 5.65 | 57.38 | 57.38 | 4.01 | 42.74 | -9.51 |
| digital | 41.53 | 39.80 | 4.14 | 31.05 | 31.05 | 0.34 | 23.44 | 11.09 |
| noise | 58.43 | 55.04 | 4.14 | 51.24 | 51.24 | 1.01 | 18.13 | 5.06 |
| weather | 43.84 | 36.16 | 7.80 | 41.63 | 41.63 | 4.32 | 17.80 | 10.91 |
| Fig. 3 (iii) | | | | | | | | |
| blur | 57.10 | 69.84 | 13.43 | 74.01 | 74.01 | 3.96 | 43.50 | 5.93 |
| digital | 46.16 | 38.06 | 12.97 | 36.22 | 36.22 | 10.52 | 4.94 | 32.01 |
| noise | 93.60 | 85.84 | 13.08 | 81.10 | 81.10 | 3.52 | 22.56 | 23.65 |
| weather | 43.74 | 36.90 | 8.98 | 44.05 | 44.05 | 6.20 | 23.29 | 3.87 |

Table 11: Estimating top-1 error of unseen corruptions within the different corruption classes. We note that especially for well defined corruptions (like noise or digital corruptions), the estimation scheme works well. We follow the categorization originally proposed by Hendrycks and Dietterich [2].

## C.6 Training details on the models trained with Fixup initialization and GroupNorm

In Section 5 of the main paper, we consider IN models trained with GroupNorm and Fixup initialization. For these models, we consider the original reference implementations provided by the authors. We train ResNet-50, ResNet-101 and ResNet-152 models with stochastic gradient descent with momentum (learning rate 0.1, momentum 0.9), with batch size 256 and weight decay $1 \times 10^{-4}$ for 100 epochs.

## C.7 Effect of Pseudo Batchsize

We show the full results for considering different choices of $N$ for ResNet-50, Augmix, ANT, ANT+SIN and SIN models and display the result in Fig. 12. We observe a characteristic shape which we believe can be attributed to the way statistics are estimated. We provide evidence for this view by proposing an analytical model which we discuss in §D.

Figure 11: Left: Performance for all the considered ResNet-50 variants based on the sample batch size. The optimal $N$ is chosen according to the mCE on the holdout corruptions. Right: Best choice for $N$ depending on the input batchsize $n$. Note that in general for high values $n$, the model is generally more robust to the choice of $N$.

Figure 12: Effects of batch size $n$ and pseudo batch size $N$ for the various considered models. We report mCE averaged across 15 test corruptions.

| ResNet-50 | 1 | 2 | 4 | 8 | 16 | 32 | 64 | 128 | 256 |
|---|---|---|---|---|---|---|---|---|---|
| 1 | 117.76 | 98.78 | 81.06 | 72.80 | 71.39 | 72.72 | 74.28 | 75.36 | 75.99 |
| 2 | 98.11 | 89.92 | 80.13 | 72.36 | 69.63 | 70.39 | 72.39 | 74.16 | 75.32 |
| 4 | 81.10 | 78.45 | 74.70 | 70.27 | 67.48 | 67.69 | 69.77 | 72.19 | 74.10 |
| 8 | 71.56 | 70.74 | 69.44 | 67.56 | 65.60 | 65.02 | 66.70 | 69.41 | 72.07 |
| 16 | 66.82 | 66.52 | 66.06 | 65.32 | 64.29 | 63.32 | 63.81 | 66.19 | 69.24 |
| 32 | 64.51 | 64.39 | 64.19 | 63.87 | 63.38 | 62.72 | 62.21 | 63.22 | 65.94 |
| 64 | 63.33 | 63.28 | 63.19 | 63.05 | 62.81 | 62.43 | 61.95 | 61.68 | 62.90 |
| 128 | 62.78 | 62.75 | 62.69 | 62.62 | 62.50 | 62.29 | 62.00 | 61.56 | 61.42 |
| 256 | 62.51 | 62.49 | 62.44 | 62.41 | 62.32 | 62.22 | 62.01 | 61.73 | 61.35 |
| 512 | 62.36 | 62.36 | 62.33 | 62.29 | 62.26 | 62.17 | 62.06 | 61.90 | 61.62 |

| AugMix | 1 | 2 | 4 | 8 | 16 | 32 | 64 | 128 | 256 |
|---|---|---|---|---|---|---|---|---|---|
| 1 | 122.56 | 99.72 | 76.23 | 65.46 | 62.08 | 61.78 | 62.70 | 63.75 | 64.47 |
| 2 | 100.39 | 88.69 | 75.16 | 64.86 | 60.93 | 60.51 | 61.28 | 62.52 | 63.67 |
| 4 | 78.55 | 74.41 | 68.69 | 62.52 | 58.58 | 58.30 | 59.53 | 60.94 | 62.39 |
| 8 | 65.02 | 63.81 | 61.86 | 59.21 | 56.39 | 55.40 | 56.87 | 59.00 | 60.77 |
| 16 | 58.02 | 57.55 | 56.96 | 56.02 | 54.69 | 53.44 | 53.78 | 56.15 | 58.71 |
| 32 | 54.37 | 54.20 | 53.99 | 53.68 | 53.21 | 52.50 | 51.99 | 53.01 | 55.78 |
| 64 | 52.55 | 52.50 | 52.38 | 52.24 | 52.07 | 51.83 | 51.39 | 51.25 | 52.59 |
| 128 | 51.64 | 51.60 | 51.54 | 51.47 | 51.38 | 51.26 | 51.10 | 50.88 | 50.89 |
| 256 | 51.18 | 51.17 | 51.12 | 51.08 | 51.02 | 50.95 | 50.86 | 50.76 | 50.60 |
| 512 | 50.96 | 50.95 | 50.93 | 50.90 | 50.86 | 50.80 | 50.72 | 50.65 | 50.61 |

| ANT | 1 | 2 | 4 | 8 | 16 | 32 | 64 | 128 | 256 |
|---|---|---|---|---|---|---|---|---|---|
| 1 | 116.10 | 93.58 | 72.31 | 62.28 | 60.07 | 60.73 | 61.75 | 62.48 | 62.90 |
| 2 | 93.88 | 83.74 | 72.01 | 62.69 | 58.97 | 59.10 | 60.44 | 61.67 | 62.44 |
| 4 | 74.51 | 71.06 | 66.34 | 61.15 | 57.55 | 57.03 | 58.51 | 60.29 | 61.64 |
| 8 | 63.65 | 62.50 | 60.74 | 58.43 | 56.04 | 55.02 | 56.10 | 58.22 | 60.20 |
| 16 | 58.37 | 57.87 | 57.14 | 56.11 | 54.77 | 53.67 | 53.76 | 55.61 | 58.06 |
| 32 | 55.78 | 55.54 | 55.20 | 54.66 | 53.91 | 53.06 | 52.50 | 53.18 | 55.35 |
| 64 | 54.51 | 54.41 | 54.21 | 53.88 | 53.42 | 52.84 | 52.23 | 51.94 | 52.87 |
| 128 | 53.92 | 53.85 | 53.71 | 53.53 | 53.28 | 52.85 | 52.29 | 51.80 | 51.65 |
| 256 | 53.66 | 53.61 | 53.50 | 53.37 | 53.20 | 52.96 | 52.54 | 52.04 | 51.60 |
| 512 | 53.53 | 53.49 | 53.41 | 53.33 | 53.21 | 53.02 | 52.78 | 52.38 | 51.90 |

| ANT+SIN | 1 | 2 | 4 | 8 | 16 | 32 | 64 | 128 | 256 |
|---|---|---|---|---|---|---|---|---|---|
| 1 | 108.24 | 84.75 | 67.42 | 59.91 | 58.15 | 58.49 | 59.24 | 59.85 | 60.23 |
| 2 | 87.60 | 78.40 | 68.32 | 60.63 | 57.54 | 57.47 | 58.33 | 59.23 | 59.87 |
| 4 | 71.12 | 68.32 | 64.31 | 59.78 | 56.63 | 56.06 | 57.01 | 58.24 | 59.23 |
| 8 | 62.23 | 61.38 | 59.98 | 57.93 | 55.69 | 54.59 | 55.30 | 56.79 | 58.21 |
| 16 | 57.83 | 57.51 | 57.00 | 56.17 | 54.96 | 53.76 | 53.61 | 54.92 | 56.68 |
| 32 | 55.62 | 55.51 | 55.33 | 54.96 | 54.38 | 53.55 | 52.80 | 53.13 | 54.73 |
| 64 | 54.57 | 54.49 | 54.40 | 54.25 | 53.98 | 53.51 | 52.84 | 52.36 | 52.89 |
| 128 | 54.02 | 53.98 | 53.95 | 53.85 | 53.72 | 53.49 | 53.07 | 52.53 | 52.12 |
| 256 | 53.76 | 53.74 | 53.71 | 53.67 | 53.59 | 53.47 | 53.23 | 52.85 | 52.33 |
| 512 | 53.64 | 53.63 | 53.60 | 53.57 | 53.51 | 53.45 | 53.35 | 53.12 | 52.75 |

| SIN | 1 | 2 | 4 | 8 | 16 | 32 | 64 | 128 | 256 |
|---|---|---|---|---|---|---|---|---|---|
| 1 | 119.11 | 94.43 | 74.93 | 67.03 | 65.43 | 66.08 | 67.16 | 68.04 | 68.62 |
| 2 | 98.85 | 88.62 | 76.99 | 67.88 | 64.23 | 64.42 | 65.72 | 67.02 | 67.99 |
| 4 | 81.35 | 78.10 | 73.38 | 67.84 | 63.49 | 62.47 | 63.76 | 65.48 | 66.94 |
| 8 | 70.92 | 69.94 | 68.38 | 66.02 | 63.14 | 61.09 | 61.45 | 63.35 | 65.35 |
| 16 | 65.29 | 64.97 | 64.48 | 63.68 | 62.39 | 60.78 | 59.90 | 60.92 | 63.16 |
| 32 | 62.34 | 62.25 | 62.08 | 61.80 | 61.36 | 60.55 | 59.55 | 59.26 | 60.65 |
| 64 | 60.84 | 60.80 | 60.74 | 60.61 | 60.47 | 60.15 | 59.67 | 58.96 | 58.93 |
| 128 | 60.07 | 60.04 | 60.02 | 59.96 | 59.87 | 59.77 | 59.57 | 59.18 | 58.64 |
| 256 | 59.68 | 59.66 | 59.64 | 59.62 | 59.59 | 59.53 | 59.43 | 59.27 | 58.97 |
| 512 | 59.48 | 59.47 | 59.46 | 59.44 | 59.42 | 59.40 | 59.33 | 59.26 | 59.11 |

| DeepAugment | 1 | 2 | 4 | 8 | 16 | 32 | 64 | 128 | 256 |
|---|---|---|---|---|---|---|---|---|---|
| 8 | 65.37 | 63.87 | 61.37 | 58.11 | 54.48 | 52.17 | 52.33 | 54.18 | 56.36 |

| DeepAugment+AugMix | 1 | 2 | 4 | 8 | 16 | 32 | 64 | 128 | 256 |
|---|---|---|---|---|---|---|---|---|---|
| 8 | 52.59 | 51.98 | 51.05 | 49.83 | 48.5 | 47.81 | 48.36 | 49.72 | 51.12 |

| ResNex+DeepAugment+Augmix | 1 | 2 | 4 | 8 | 16 | 32 | 64 | 128 | 256 |
|---|---|---|---|---|---|---|---|---|---|
| 8 | 42.09 | 41.74 | 41.29 | 40.67 | 39.96 | 39.69 | 40.35 | 41.55 | 42.69 |

Table 12: Test mCE for various batch sizes (rows) vs. pseudo batch sizes (columns)

# D  Analytical error model

We consider a univariate model in §D.1–D.3 and discuss a simple extension to the multivariate diagonal case in §D.4. As highlighted in the main text, the model qualitatively explains the overall characteristics of our experimental data. Note that we assume a linear relationship between the Wasserstein distance and the error under domain shift, as suggested by our empirical findings.

**Univariate model.**  We denote the source statistics as $\mu_s, \sigma_s^2$, the true target statistics as $\mu_t, \sigma_t^2$ and the estimated target statistics as $\hat{\mu}_t, \hat{\sigma}_t^2$. For normalization, we take a convex combination of the source statistics and estimated target statistics:

$$\bar{\mu} = \frac{N}{N+n}\mu_s + \frac{n}{N+n}\hat{\mu}_t, \ \bar{\sigma}^2 = \frac{N}{N+n}\sigma_s^2 + \frac{n}{N+n}\hat{\sigma}_t^2. \tag{11}$$

We now analyze the trade-off between using an estimate closer to the source or closer to the estimated target statistics. In the former case, the model will suffer under the covariate shift present between target and source distribution. In the latter case, small batch sizes $n$ will yield unreliable estimates for the true target statistics, which might hurt the performance even more than the source-target mismatch. Hence, we aim to gain understanding in the trade-off between both options, and potential optimal choices of $N$ for a given sample size $n$.

As a metric of domain shift with good properties for our following derivation, we leverage the Wasserstein distance. In §5 and §C.5, we already established an empirical link between domain shift measured in terms of the top-1 performance vs. the Wasserstein distance between model statistics and observed a linear relationship for case of common corruptions.

**Proposition 1** (Bounds on the expected value of the Wasserstein distance between target and combined estimated target and source statistics). *We denote the source statistics as $\mu_s, \sigma_s^2$, the true target statistics as $\mu_t, \sigma_t^2$ and the biased estimates of the target statistics as $\hat{\mu}_t, \hat{\sigma}_t^2$. For normalization, we take a convex combination of the source statistics and estimated target statistics as discussed in Eq. 11. At a confidence level $1 - \alpha$, the expectation value of the squared Wasserstein distance $W_2^2(\bar{\mu}, \bar{\sigma}, \mu_t, \sigma_t)$ between ideal and estimated target statistics w.r.t. to the distribution of sample mean $\hat{\mu}_t$ and sample variance $\hat{\sigma}_t^2$ is bounded from above and below with $L \le \mathbb{E}[W_2^2] \le U$, where*

$$L = \left( \sigma_t - \sqrt{\frac{N}{N+n}\sigma_s^2 + \frac{n-1}{N+n}\sigma_t^2} \right)^2 + \frac{N^2}{(N+n)^2}(\mu_t - \mu_s)^2 + \frac{n}{(N+n)^2}\sigma_t^2$$

$$U = \ L + \sigma_t^5 \frac{(n-1)}{2(N+n)^2}\left( \frac{N}{N+n}\sigma_s^2 + \frac{1}{N+n}\chi_{1-\alpha/2,n-1}^2\sigma_t^2 \right)^{-3/2} \tag{12}$$

*The quantity $\chi_{1-\alpha/2,n-1}^2$ denotes the left tail value of a chi square distribution with $n-1$ degrees of freedom, defined as $P\left( X \le \chi_{1-\alpha/2,n-1}^2 \right) = \alpha/2$ for $X \sim \chi_{n-1}^2$.*

## D.1  Proof sketch

We are interested in the expected value of the Wasserstein distance defined in (A.1) between the target statistics $\mu_t, \sigma_t^2$ and the mixed statistics $\bar{\mu}, \bar{\sigma}^2$ introduced above in equation (11), taken with respect to the distribution of the sample moments $\hat{\mu}_t, \hat{\sigma}_t^2$. The expectation value itself cannot be evaluated in closed form because the Wasserstein distance contains a term proportional to $\bar{\sigma}$ being the square root of the convex combination of target and source variance.

In Lemma 3, the square root term is bounded from above and below using Jensen's inequality and Holder's defect formula which is reviewed in Lemma 2. After having bounded the problematic square root term, the proof of Proposition 1 reduces to inserting the expectation values of sample mean and sample variance reviewed in Lemma 1.

## D.2 Prerequisites

**Lemma 1** (Mean and variance of sample moments, following [65]). *The sample moments $\hat{\mu}_t, \hat{\sigma}_t^2$ are random variables depending on the sample size $n$.*

$$\hat{\mu}_t = \frac{1}{n}\sum_{j=1}^n x_j, \quad \hat{\sigma}_t^2 = \frac{1}{n}\sum_{j=1}^n (x_j - \hat{\mu}_t)^2 \ \text{ with } x_j \sim \mathcal{N}\left(\mu_t, \sigma_t^2\right). \tag{13}$$

*For brevity, we use the shorthand $\mathbb{E}[\cdot]$ for all expectation values with respect to the distribution of $p(\hat{\mu}_t, \hat{\sigma}_t^2 | n)$. In particular, our computation uses mean and variance of $\hat{\mu}_t$ and $\hat{\sigma}_t^2$ which are well known for a normal target distribution:*

$$\hat{\mu}_t \sim \mathcal{N}\left(\mu_t, \frac{1}{n}\sigma_t^2\right), \ \mathbb{E}[\hat{\mu}_t] = \mu_t, \ \mathbb{V}[\hat{\mu}_t] = \frac{1}{n}\sigma_t^2 \tag{14}$$

$$\frac{\hat{\sigma}_t^2}{\sigma_t^2/n} \sim \chi_{n-1}^2, \ \mathbb{E}[\hat{\sigma}_t^2] = \frac{n-1}{n}\sigma_t^2, \ \mathbb{V}[\hat{\sigma}_t^2] = \frac{\sigma_t^4}{n^2}\mathbb{V}\left[\frac{\hat{\sigma}_t^2}{\sigma_t^2/n}\right] = \frac{\sigma_t^4}{n^2}2(n-1). \tag{15}$$

*The derivation of the variance $\mathbb{V}[\hat{\sigma}_t^2]$ in the last line uses the fact that the variance of a chi square distributed variable with $(n-1)$ degrees of freedom is equal to $2(n-1)$.*

**Lemma 2** (Holder's defect formula for concave functions in probabilistic notation, following Becker [66] ). *If the concave function $f : [a, b] \to \mathbb{R}$ is twice continuously differentiable and there are finite bounds $m$ and $M$ such that*

$$-M \leq f''(x) \leq -m \leq 0 \ \forall x \in [a, b], \tag{16}$$

*then the defect between Jensen's inequality estimate $f\left(\mathbb{E}[X]\right)$ for a random variable $X$ taking values $x \in [a, b]$ and the true expectation value $\mathbb{E}[f(X)]$ is bounded from above by a term proportional to the variance of $X$:*

$$f\left(\mathbb{E}[X]\right) - \mathbb{E}[f(X)] \leq \frac{1}{2}M\mathbb{V}[X]. \tag{17}$$

**Lemma 3** (Upper and lower bounds on the expectation value of $\bar{\sigma}$). *The expectation value of the square root of the random variable $\bar{\sigma}^2$ defined as*

$$\bar{\sigma}^2 = \frac{N}{N+n}\sigma_s^2 + \frac{n}{N+n}\hat{\sigma}_t^2, \tag{18}$$

*is bounded from above and below at a confidence level $1 - \alpha$ by*

$$\sqrt{\mathbb{E}\left[\bar{\sigma}^2\right]} - \frac{1}{2}M\mathbb{V}[\bar{\sigma}^2] \leq \mathbb{E}\left[\sqrt{\bar{\sigma}^2}\right] \leq \sqrt{\mathbb{E}\left[\bar{\sigma}^2\right]} \tag{19}$$

$$\sqrt{\mathbb{E}\left[\bar{\sigma}^2\right]} = \sqrt{\frac{N}{N+n}\sigma_s^2 + \frac{n-1}{N+n}\sigma_t^2}, \tag{20}$$

$$\frac{1}{2}M\mathbb{V}[\bar{\sigma}^2] = \frac{(n-1)}{4(N+n)^2}\sigma_t^4\left(\frac{N}{N+n}\sigma_s^2 + \frac{1}{N+n}\chi_{1-\alpha/2,n-1}^2\sigma_t^2\right). \tag{21}$$

*The quantity $\chi_{1-\alpha/2,n-1}^2$ denotes the left tail value of a chi square distribution with $n-1$ degrees of freedom, defined as $P\left(X \leq \chi_{1-\alpha/2,n-1}^2\right) = \alpha/2$ for $X \sim \chi_{n-1}^2$.*

*Proof.* The square root function is concave, therefore Jensen's inequality implies the upper bound

$$\mathbb{E}\left[\sqrt{\bar{\sigma}^2}\right] \leq \sqrt{\mathbb{E}[\bar{\sigma}^2]}. \tag{22}$$

The square root of the expectation value of $\bar{\sigma}^2$ is computed using the expectation value of the sample variance as given in Lemma 1.

$$\sqrt{\mathbb{E}[\bar{\sigma}^2]} = \sqrt{\frac{N}{N+n}\sigma_s^2 + \frac{n}{N+n}\frac{n-1}{n}\sigma_t^2} = \sqrt{\frac{N}{N+n}\sigma_s^2 + \frac{n-1}{N+n}\sigma_t^2}. \tag{23}$$

To state a lower bound, we use Holder's defect formula in probabilistic notation stated in Lemma 2. Holder's formula for concave functions requires that the random variable $\bar{\sigma}^2$ can take values in

the compact interval $[a, b]$ and that the second derivative of the square root function $f(\bar{\sigma}^2) = \sqrt{\bar{\sigma}^2}$, exists and is strictly smaller than zero in $[a, b]$. Regarding the interval of $\bar{\sigma}^2$, we provide probabilistic upper and lower bounds. The ratio of sample variance and true variance divided by $n$ follows a chi square distribution with $n - 1$ degrees of freedom. At confidence level $1 - \alpha$, this ratio lies between $\chi^2_{1-\alpha/2,n-1}$ and $\chi^2_{\alpha/2,n-1}$ which are defined as follows:

$$\chi^2_{1-\alpha/2,n-1} \leq \frac{\hat{\sigma}_t^2}{\sigma_t^2/n} \leq \chi^2_{\alpha/2,n-1}, \tag{24}$$

$$Pr(X \leq \chi^2_{1-\alpha/2,n-1}) = \frac{\alpha}{2}, \ Pr(X \geq \chi^2_{\alpha/2,n-1}) = \frac{\alpha}{2}. \tag{25}$$

Then at the same confidence level, the sample variance itself lies between the two quantiles multiplied by $\sigma_t^2/n$,

$$\chi^2_{1-\alpha/2,n-1}\frac{\sigma_t^2}{n} \leq \hat{\sigma}_t^2 \leq \chi^2_{\alpha/2,n-1}\frac{\sigma_t^2}{n}, \tag{26}$$

and the random variable $\bar{\sigma}^2$ lies in the interval

$$\bar{\sigma}^2 \in [a, b] \text{ with } a = \frac{N}{N+n}\sigma_s^2 + \frac{1}{N+n}\chi^2_{1-\alpha/2,n-1}\sigma_t^2, \tag{27}$$

$$\text{and } b = \frac{N}{N+n}\sigma_s^2 + \frac{1}{N+n}\chi^2_{\alpha/2,n-1}\sigma_t^2. \tag{28}$$

The variances and chi square values are all positive and therefore both $a$ and $b$ are positive as well, implying that the second derivative of the square root is strictly negative in the interval $[a, b]$.

$$f(\bar{\sigma}^2) = \sqrt{\bar{\sigma}^2}, \ f'(\bar{\sigma}^2) = \frac{1}{2}(\bar{\sigma}^2)^{-1/2}, \ f''(\bar{\sigma}^2) = -\frac{1}{4}(\bar{\sigma}^2)^{-3/2} < 0 \in [a, b]. \tag{29}$$

Consequently the second derivative is in the interval $[M, m]$ at the given confidence level:

$$-M \leq f''(\bar{\sigma}^2) \leq -m \leq 0 \text{ for } \bar{\sigma}^2 \in [a, b] \text{ with } M = \frac{1}{4}a^{-3/2}, \ m = \frac{1}{4}b^{-3/2}. \tag{30}$$

The defect formula 2 states that the defect is bounded by

$$\sqrt{\mathbb{E}[\bar{\sigma}^2]} - \mathbb{E}[\sqrt{\bar{\sigma}^2}] \leq \frac{1}{2}M\mathbb{V}[\bar{\sigma}^2]. \tag{31}$$

The constant $M$ was computed above in (30), and the variance of $\bar{\sigma}^2$ is calculated in the next lines, using the first and second moment of the sample variance as stated in 1.

$$\mathbb{V}[\bar{\sigma}^2] = \mathbb{E}[(\bar{\sigma}^2 - \mathbb{E}[\bar{\sigma}^2])^2] = \mathbb{E}\left[\left(\frac{n}{N+n}\hat{\sigma}_t^2 - \frac{n}{N+n}\frac{n-1}{n}\sigma_t^2\right)^2\right]$$

$$= \frac{n^2}{(N+n)^2}\mathbb{E}\left[(\hat{\sigma}_t^2 - \mathbb{E}[\hat{\sigma}_t^2])^2\right] = \frac{n^2}{(N+n)^2}\mathbb{V}[\hat{\sigma}_t^2] \tag{32}$$

$$= \frac{n^2}{(N+n)^2}\frac{2(n-1)}{n^2}\sigma_t^4 = \frac{2(n-1)}{(N+n)^2}\sigma_t^4.$$

Inserting $\mathbb{V}[\bar{\sigma}^2]$ computed in (32) and $M$ defined in (30) with $a$ as defined in (27) into the defect formula (31) yields the lower bound:

$$\sqrt{\mathbb{E}[\bar{\sigma}^2]} - \frac{1}{2}M\mathbb{V}[\bar{\sigma}^2] \leq \mathbb{E}[\sqrt{\bar{\sigma}^2}]$$

$$\sqrt{\mathbb{E}[\bar{\sigma}^2]} - \frac{1}{2}M\mathbb{V}[\bar{\sigma}^2]$$

$$= \sqrt{\mathbb{E}[\bar{\sigma}^2]} - \frac{1}{2}\cdot\frac{1}{4}a^{-3/2}\frac{2(n-1)}{(N+n)^2}\sigma_t^4 \tag{33}$$

$$= \sqrt{\mathbb{E}[\bar{\sigma}^2]} - \frac{(n-1)}{4(N+n)^2}\sigma_t^4\left(\frac{N}{N+n}\sigma_s^2 + \frac{1}{N+n}\chi^2_{1-\alpha/2,n-1}\sigma_t^2\right)^{-3/2}.$$

Assuming that source and target variance are of the same order of magnitude $\sigma$, the defect will be of order of magnitude $\sigma$: The factor $\mathbb{V}[X]$ scales with $\sigma^4$ and $M$ with $\sigma^{-3}$. $\qquad\square$

### D.3 Proof of Proposition 1

*Proof.* For two univariate normal distributions with moments $\mu_t, \sigma_t^2$ and $\bar{\mu}, \bar{\sigma}^2$, the Wasserstein distance as defined in (A.1) reduces to

$$W_2^2 = \sigma_t^2 + \bar{\sigma}^2 - 2\bar{\sigma}\sigma_t + (\bar{\mu} - \mu)^2. \tag{34}$$

The expected value of the Wasserstein distance across many batches is given as

$$\begin{aligned}
\mathbb{E}[W_2^2] &= \sigma_t^2 + \mathbb{E}[\bar{\sigma}^2] - 2\mathbb{E}[\bar{\sigma}]\sigma_t + \mathbb{E}[(\mu_t - \bar{\mu})^2] \\
&= \sigma_t^2 + \frac{N}{N+n}\sigma_s^2 + \frac{n}{N+n}\frac{n-1}{n}\sigma_t^2 - 2\sigma_t\mathbb{E}\left[\sqrt{\frac{N}{N+n}\sigma_s^2 + \frac{n}{N+n}\hat{\sigma}_t^2}\right] \\
&\quad + \mathbb{E}\left[\left(\mu_t - \frac{N}{N+n}\mu_s - \frac{n}{N+n}\hat{\mu}_t\right)^2\right]
\end{aligned} \tag{35}$$

which can already serve as the basis for our numerical simulations. To arrive at a closed form analytical solution, we invoke Lemma 3 to bound the expectation value $\mathbb{E}[\bar{\sigma}]$ in equation (35).

$$-2\sigma_t\sqrt{\mathbb{E}[\bar{\sigma}^2]} \leq -2\sigma_t\mathbb{E}\left[\sqrt{\bar{\sigma}^2}\right] \leq -2\sigma_t\sqrt{\mathbb{E}[\bar{\sigma}^2]} - 2\sigma_t\left(-\frac{1}{2}M\mathbb{V}[\bar{\sigma}^2]\right) \tag{36}$$

Apart from the square root term bounded in equation (36) above, the expectation value of the Wasserstein distance can be computed exactly. Hence the bounds on $\mathbb{E}[\bar{\sigma}]$ multiplied by a factor of $(-2\sigma_t^2)$ coming from equation (35) determine lower and upper bounds $L$ and $U$ on the expected value of $W_2^2$:

$$L \leq \mathbb{E}\left[W_2^2\right] \leq U = L + \sigma_t M\mathbb{V}[\bar{\sigma}^2] \tag{37}$$

In the next lines, the lower bound is calculated:

$$\begin{aligned}
L = {}& \sigma_t^2 + \frac{N}{N+n}\sigma_s^2 + \frac{n-1}{N+n}\sigma_t^2 - 2\sigma_t\sqrt{\mathbb{E}\left[\frac{N}{N+n}\sigma_s^2 + \frac{n-1}{N+n}\sigma_t^2\right]} \\
&+ \left(\mu_t - \frac{N}{N+n}\mu_s\right)^2 - 2\left(\mu_t - \frac{N}{N+n}\mu_s\right)\frac{n}{N+n}\mathbb{E}[\hat{\mu}_t] + \frac{n^2}{(N+n)^2}\left(\mathbb{V}[\hat{\mu}_t] + (\mathbb{E}[\hat{\mu}_t])^2\right) \\
= {}& \sigma_t^2 + \frac{N}{N+n}\sigma_s^2 + \frac{n-1}{N+n}\sigma_t^2 - 2\sigma_t\sqrt{\frac{N}{N+n}\sigma_s^2 + \frac{n-1}{N+n}\sigma_t^2} \\
&+ \left(\mu_t - \frac{N}{N+n}\mu_s\right)^2 - 2\left(\mu_t - \frac{N}{N+n}\mu_s\right)\frac{n}{N+n}\mu_t + \frac{n^2}{(N+n)^2}\left(\frac{1}{n}\sigma_t^2 + \mu_t^2\right) \\
= {}& \left(\sigma_t - \sqrt{\frac{N}{N+n}\sigma_s^2 + \frac{n-1}{N+n}\sigma_t^2}\right)^2 + \left(\mu_t - \frac{N}{N+n}\mu_s - \frac{n}{N+n}\mu_t\right)^2 + \frac{n}{(N+n)^2}\sigma_t^2 \\
= {}& \left(\sigma_t - \sqrt{\frac{N}{N+n}\sigma_s^2 + \frac{n-1}{N+n}\sigma_t^2}\right)^2 + \frac{N^2}{(N+n)^2}(\mu_t - \mu_s)^2 + \frac{n}{(N+n)^2}\sigma_t^2
\end{aligned} \tag{38}$$

After having derived the lower bound, the upper bound is the sum of the lower bound and the defect term as computed in Lemma 3.

$$\begin{aligned}
\mathbb{E}[W^2] \geq U &= L + \sigma_t M\mathbb{V}[\bar{\sigma}^2] \\
&= L + \sigma_t\frac{1}{4}\left(\frac{N}{N+n}\sigma_s^2 + \frac{n}{N+n}\chi_{1-\alpha/2,n-1}^2\frac{\sigma_t^2}{n}\right)^{-3/2}\frac{2(n-1)}{(N+n)^2}\sigma_t^4 \tag{39} \\
&= L + \left(\frac{N}{N+n}\sigma_s^2 + \frac{1}{N+n}\chi_{1-\alpha/2,n-1}^2\sigma_t^2\right)^{-3/2}\frac{(n-1)}{2(N+n)^2}\sigma_t^5.
\end{aligned}$$

$\square$

Based on choices of the model parameters, the model qualitatively matches our experimental results. We plot different choices in Fig. 13.

Figure 13: Overview of different parametrizations of the model. We denote each plot with $(\mu_t - \mu_s, \sigma_t/\sigma_s)$ and report the lower bound $\sqrt{L}$ on the Wasserstein distance. Parametrizations in columns four to seven produce qualitatively similar results we observed in our experiments, assuming a linear relationship between the Wasserstein distance and the error rate.

## D.4 Extension to multivariate distributions.

We now derive a multivariate variant that can be fit to data from a DNN. Due to the estimation of running statistics in the network, we have access to a diagonal approximation of the true covariance matrix.

We denote the diagonal covariance matrices with matrix elements $\sigma_i^2$ as

$$(\Sigma_t)_{ii} = (\sigma_t^2)_i, \ (\hat{\Sigma}_t)_{ii} = (\hat{\sigma}_t^2)_i, \ (\Sigma_s)_{ii} = (\sigma_s^2)_i \tag{40}$$

and extend our definition of the statistics used for normalization to $\bar{\mu}$ and $\bar{\Sigma}$:

$$\bar{\mu} = \frac{N}{N+n}\mu_s + \frac{n}{N+n}\hat{\mu}_t, \ \bar{\Sigma} = \frac{N}{N+n}\Sigma_s + \frac{n}{N+n}\hat{\Sigma}_t. \tag{41}$$

The Wasserstein distance between $\bar{\mu}, \bar{\Sigma}$ and $\mu_t, \Sigma_t$ is then defined as

$$\begin{aligned}
W_2^2 &= \operatorname{Tr}\Sigma_t + \bar{\Sigma} - 2\Sigma_t^{1/2}\bar{\Sigma}^{1/2} + (\mu_t - \bar{\mu})^T(\mu_t - \bar{\mu}) \\
&= \sum_{i=1}^{D}(\sigma_t^2)_i + (\bar{\sigma}^2)_i - 2(\bar{\sigma})_i(\sigma_t)_i + ((\mu_t)_i - (\bar{\mu}_t)_i)^2 = \sum_{i=1}^{D}(W_2^2)_i
\end{aligned} \tag{42}$$

Every component $(W_2^2)_i$ in the sum above is bounded by the univariate bound discussed above. The multivariate Wasserstein distance which sums over the diagonal covariance matrix entries is then bounded by the sums over the individual bounds $L_i$ and $U_i$ given in (12).

$$L_i \le (W_2^2)_i \le U_i \Rightarrow \sum_{i=1}^{D} L_i \le W_2^2 \le \sum_{i=1}^{D} U_i. \tag{43}$$

## D.5 Limits of Proposition 1

**Limit** $n \to \infty$ In the limit of infinite batch size $n \to \infty$, upper and lower bounds on the expected Wasserstein distance between $\bar{\mu}, \bar{\sigma}^2$ and $\mu_t, \sigma_t^2$ both go to zero.

$$\begin{aligned}
\lim_{n\to\infty} L &= \lim_{n\to\infty}\left(\sigma_t - \sqrt{\frac{N}{N+n}\sigma_s^2 + \frac{n-1}{N+n}\sigma_t^2}\right)^2 + \frac{N^2}{(N+n)^2}(\mu_t - \mu_s)^2 + \frac{n}{(N+n)^2}\sigma_t^2 \\
&= (\sigma_t - \sigma_t)^2 = 0 \\
\lim_{n\to\infty} U &= \lim_{n\to\infty} L + \lim_{n\to\infty}\sigma_t^5\frac{(n-1)}{2(N+n)^2}\left(\frac{N}{N+n}\sigma_s^2 + \frac{1}{N+n}\chi_{1-\alpha/2,n-1}^2\sigma_t^2\right)^{-3/2} = 0.
\end{aligned} \tag{44}$$

The intuition behind this limit is that if a large number of samples from the target domain is given, $\hat{\mu}$ and $\hat{\sigma}^2$ approximate the true target statistics very well. As $\hat{\mu}$ and $\hat{\sigma}^2$ dominate $\bar{\mu}$ and $\bar{\sigma}^2$ for large $n$, the expected Wasserstein distance has to vanish.

**Limit** $N \to \infty$ In the opposite limit $N \to \infty$, the expected value of the Wasserstein distance reduces to the Wasserstein distance between source and target statistics.

$$\lim_{N\to\infty} \bar{\mu} = \mu_s, \ \lim_{N\to\infty} \bar{\sigma}^2 = \sigma_s^2, \tag{45}$$

$$\Rightarrow \lim_{N\to\infty} \mathbb{E}[W_2^2] = \sigma_t^2 + \sigma_s^2 - 2\sigma_t\sigma_s + (\mu_t - \mu_s)^2 = W_2^2\left(\mu_s, \sigma_s^2, \mu_t, \sigma_t^2\right). \tag{46}$$

**Limiting case** $\mu_t = \mu_s$ **and** $\sigma_t^2 = \sigma_s^2$ When source and target domain coincide, and the statistics $\sigma_s^2 = \sigma_t^2$ and $\mu_s = \mu_t$ are known, then the source target mismatch is not an error source.

However, one might assume that source and target domain are different even though they actually coincide. In this case, proceeding with our proposed strategy and using the statistics $\bar{\mu}$ and $\bar{\sigma}^2$, the bounds on the expected Wasserstein distance follow from setting $\sigma_t^2$ to $\sigma_s^2$ and $\mu_t$ to $\mu_s$ in

Proposition 1.

$$\bar{\mu} = \frac{N}{N+n}\mu_t + \frac{n}{N+n}\hat{\mu}_t, \ \bar{\sigma}^2 = \frac{N}{N+n}\sigma_t^2 + \frac{n}{N+n}\hat{\sigma}_t^2, \ L \leq \mathbb{E}[W_2^2] \leq U$$

$$L = \sigma_t^2 \left( \frac{2N^2 + 4Nn - N + 2n^2}{(N+n)^2} - 2\sqrt{1 - \frac{1}{N+n}} \right),$$

$$U = L + \sigma_t^2 \frac{n-1}{2(N+n)^2} \left( \frac{N + \chi_{1-\alpha/2,n-1}^2}{N+n} \right)^{-3/2}. \tag{47}$$

It could also be the case that the equality of source and target statistics is known but the concrete values of the statistics are unknown. In our model, this amounts to setting the number of pseudo samples $N$ to zero and assuming that source and target statistics are equal. Setting $N = 0$ in equation (47) and keeping $n$ finite yields

$$L = 2\sigma_t^2 \left( 1 - \sqrt{1 - \frac{1}{n}} \right), \ U = L + \sigma_t^2 \frac{n-1}{2n^2} \left( \frac{\chi_{1-\alpha/2,n-1}^2}{n} \right)^{-3/2}. \tag{48}$$

## D.6  Bounds on the normalized Wasserstein distance

The Wasserstein distance (cf. §A.1) between the interpolating statistics $\bar{\mu}$, $\bar{\sigma}^2$ and the target statistics can also be normalized by a factor of $\sigma_s^{-2}$. Because $\sigma_s^{-2}$ is constant, the bounds on the expectation value of the unnormalized Wasserstein distance discussed in the previous subsections just have to be multiplied by $\sigma_s^{-2}$ to obtain bounds on the normalized Wasserstein distance (cf. §A.2):

$$\frac{L}{\sigma_s^2} \leq \widetilde{W}_2^2 = W_2^2 \left( \frac{\bar{\mu}}{\sigma_s}, , \frac{\bar{\sigma}^2}{\sigma_s^2}, \frac{\mu_t}{\sigma_s}, \frac{\sigma_t^2}{\sigma_s^2} \right) = \frac{1}{\sigma_s^2} W_2^2(\bar{\mu}, \bar{\sigma}^2, \mu_t, \sigma_t^2) \leq \frac{U}{\sigma_s^2}. \tag{49}$$

# E    Full list of models evaluated on IN

The following lists contains all models we evaluated on various datasets with references and links to the corresponding source code.

## E.1    Torchvision models trained on IN

Weights were taken from `https://github.com/pytorch/vision/tree/master/torchvision/models`

1. `alexnet` [67]
2. `densenet121` [15]
3. `densenet161` [15]
4. `densenet169` [15]
5. `densenet201` [15]
6. `densenet201` [15]
7. `googlenet` [16]
8. `inception_v3` [17]
9. `mnasnet0_5` [18]
10. `mnasnet1_0` [18]
11. `mobilenet_v2` [19]
12. `resnet18` [20]
13. `resnet34` [20]
14. `resnet50` [20]
15. `resnet101` [20]
16. `resnet152` [20]
17. `resnext50_32x4d` [21]
18. `resnext101_32x8d` [21]
19. `shufflenet_v2_x0_5` [22]
20. `shufflenet_v2_x1_0` [22]
21. `vgg11_bn` [23]
22. `vgg13_bn` [23]
23. `vgg16_bn` [23]
24. `vgg19_bn` [23]
25. `wide_resnet101_2` [24]
26. `wide_resnet50_2` [24]

## E.2    Robust ResNet50 models

1. `resnet50 AugMix` [30] `https://github.com/google-research/augmix`
2. `resnet50 SIN+IN` [28] `https://github.com/rgeirhos/texture-vs-shape`
3. `resnet50 ANT` [29] `https://github.com/bethgelab/game-of-noise`
4. `resnet50 ANT+SIN` [29] `https://github.com/bethgelab/game-of-noise`
5. `resnet50 DeepAugment` [36] `https://github.com/hendrycks/imagenet-r`
6. `resnet50 DeepAugment+AugMix` [36] `https://github.com/hendrycks/imagenet-r`

### E.3  SimCLRv2 models [27]

We used the checkpoints from `https://github.com/google-research/simclr` and converted them from TensorFlow to PyTorch with `https://github.com/tonylins/simclr-converter`, commit ID: 139d3cb0bd0c64b5ad32aab810e0bd0a0dddaae0.

1. `resnet50` FT100 SK=0 width=1
2. `resnet101` FT100 SK=0 width=1
3. `resnet152` FT100 SK=0 width=1

### E.4  Robust ResNext models [21]

Note that the baseline `resnext50_32x4d` model trained on ImageNet is available as part of the `torchvision` library.

1. `resnext50_32x4d WSL` [26] `https://github.com/facebookresearch/WSL-Images/blob/master/hubconf.py`
2. `resnext101_32x4d WSL` [26] `https://github.com/facebookresearch/WSL-Images/blob/master/hubconf.py`
3. `resnext101_32x8d Deepaugment+AugMix` [36] `https://github.com/hendrycks/imagenet-r`

### E.5  ResNet50 with Group Normalization [40]

Model weights and training code was taken from `https://github.com/ppwwyyxx/GroupNorm-reproduce`

1. `resnet50 GroupNorm`
2. `resnet101 GroupNorm`
3. `resnet152 GroupNorm`

### E.6  ResNet50 with Fixup initialization [39]

Model weights and training code was taken from `https://github.com/hongyi-zhang/Fixup/tree/master/imagenet`. For training, we keep all hyperparameters at their default values and note that in particular the batchsize of 256 is a sensitive parameter.

1. `resnet50 FixUp`
2. `resnet101 FixUp`
3. `resnet152 FixUp`

## Footnotes

[4]Note that for simplicity, we do not reset the statistics of the remaining $(b_i - i)$ BN layers. This could potentially be adapted in future work.