[Reviews · NeurIPS 2020]

Review 1

Summary and Contributions: This paper adjusts the batch norm statistics during test time and yields large gains on many distribution shifts. The adjustment depends on the number of examples, which is new. Evaluation is mainly on corrupted images of various types (ImageNet-C).

Strengths: Empirical evaluation is sound and definitely thorough and on large-scale images. The performance improvements are substantial and makes efficient use of previously unexploited information. The paper also shows when this technique might not be as helpful (ResNeXt-WSL). This is relevant to NeurIPS given the increased interest in OOD generalization.

Weaknesses: Using Adaptive BN for distribution shift robustness has been proposed in several _parallel_ works, though the sample-size dependent adjustment is distinct even from these parallel works.

Correctness: "Ford et al. [2] report a decrease in performance when the compressed JPEG files are used as opposed to applying the corruptions directly in memory without compression artifacts." This was overstated and because they accidentally used 299x299 images. The performance difference is small.

Clarity: The paper is well-organized and figures look fine.

Relation to Prior Work: Yes

Reproducibility: Yes

Additional Feedback: Is Eqn (4) a good way to integrate new statistics? Would it be better if the reliance on the original statistics decreases exponentially as new samples increase? (You're proposing a weighted average instead of an exponential moving average.) It would be good to mention _Identifying Statistical Bias in Dataset Replication_ in situating the ImageNet-V2 results. The appendix figure comparing mCE to # of parameters was informative. This type of high-level analysis is useful. ImageNet-R comparisons would be good to see in a camera-ready. Update: I have read the paper and appreciate the expanded results.


Review 2

Summary and Contributions: The authors posit that "common corruptions" (= ImageNet-C) to images translate to (first and second second moment) shifts in the activations for batch-norm trained models. They suggest to correct for this by estimating the batch statistics from the test set (or interpolate between those and the training ones) and confirm that this helps across several ImageNet SOTA models. They further show that the gains vanish when one uses models that have been pre-trained on large weakly supervised models. Finally, they draw a connection between the Wasserstein metric and model performance on ImageNet-C, by showing that they are correlated. The idea of adapting the batch-norm statistics under a distributional shift is not new, as the authors point out (paragraph starting on l. 249). Hence, the main contribution of this paper is the hypothesis that synthetic perturbations result in such corruptions, and the analysis of the proposed technique (that fixes only the first two moments). It's a bit unfortunate that the authors test their hypothesis on a single dataset only. Nevertheless, I believe this is an interesting technique that can be useful to construct strong baselines on this ever more important problem. After response: -- Thanks for your response! It is encouraging to see that you show benefits on another dataset, and thanks for fixing the presentation errors / typos. I wish you had touched upon the Wasserstein-related question, I was really curious about its importance for this problem.

Strengths: - Analyzing the model behavior under covariate shifts is an important problem that the NeurIPS community is excited about. - The technique is simple and shows and can be a very strong baseline for future developments. - They successfuly correct several different models to obtain SOTA results.

Weaknesses: - The authors did not try analyze if their hypothesis holds on other datasets, e.g. CIFAR-C, or any dataset that is derived using synthetic perturbations. - There are several issues with the clarity of the paper and the precision of the theoretical claims. - The technique itself to correct the BN statistics has been known. This limits the novelty as it sets the focus on the site on common corruptions, which is evaluated on a single dataset only.

Correctness: - The proposal to optimize a lower bound on the Wasserstein loss (line 227) is questionable, why not focus simply on the upper bound and use the lower only to quantify uncertainty? - Proposition 1 holds only for Gaussian distributions, the Wasserstein metric has no closed solution as a function of the first two moments only otherwise. - Equation (3) is wrong, you have to change the argument to p_s(x) and the right side rather than the left side of both conditionals. You could also define a bijection A=Covariance^{-1/2}(x-Mean) and work with x'=Ax.

Clarity: The paper itself is clearly written and easy to follow. One thing that could be improved is to note that the mean / variance scalings are not made for each layer independently, as they have cascading effects. This is clear from the appendix, but it should be stressed in the text. There are however several issues with the figures, e.g. - Figure 2 can be hard to read, I'd suggest changing the marker style to something else rather than filled-in and blank. - The colors in Figure 4 (middle) are completely incosistent. Further. please note in the paper itself why there are three ImageNet-v2 variants, it can be very confusing to some readers. Further, shouldn't the baseline match the "Matched" ImageNet-v2 variant? I'm either confused by the colors, or it's matching "Top". - Why are there two horizontal lines in Fig 4 (right), but only one in the legend? - Why is the baseline marked differently in the left and middle panels of Fig 4? - I'd spell out ResNet and AssembleNet fully in Fig 1 (or in the caption).

Relation to Prior Work: The authors positioned both the problem, current approaches, as well as their proposal for batch norm correction well with regard to prior work.

Reproducibility: Yes

Additional Feedback: - Why did you focus on the Wasserstein distance? I would be very curious how the plot looks like for other divergences (you mention Jensen-Shannon in the appendix, but not in the main text), as many have closed forms for Gaussians (or are easily estimated, given that they are 1-d distributions).


Review 3

Summary and Contributions: The key motivation of this work is the sensitivity of models to covariate shift. The main contribution of the work is around building an empirical understanding of adaptive Batch Normalization across a variety of models and datasets, in the presence of covariate shift.

Strengths: - As pointed out in S3, the core method that the paper studies is very close to prior work on adaptive Batch Norm. The additional idea of interpolating the train and test statistics is useful when the number of target samples is limited, and results suggest that is possible to adapt with a limited number of samples. - The work does a good job of making a case for the inclusion of evaluation settings where the model is allowed to adapt to the target domain. There seems to be a substantial performance difference with and without adaptation (on IN-C), and the additional complexity of evaluation with adaptation seems minor. I was able to look over the submitted code, and I think it would be an easy addition to existing benchmarks. - The empirical study is has some interesting takeaways: (i) adaptation seems to improve performance across the board for IN-C; (ii) pretraining may help mitigate internal covariate shift; (iii) BN with adaptation appears to be more robust than alternatives like GroupNorm or Fixup. Overall the simplicity of the method is appealing, and it provides a substantial improvement for little extra effort.

Weaknesses: - Since the contribution of the work is largely empirical, I think S5/S6 should be much more carefully written to explain results (see detailed comments on clarity and suggestions for improvement). - Parts of the paper appear to be somewhat uncautiously written: e.g. while lines 184-190 (correctly) suggest that pretraining alleviates the need for adaptation based on Table 2, lines 285-287 make a stronger claim about the covariate shift under the IG-3.5B model, which is not something that has been explicitly shown empirically.

Correctness: Results seem correct (errata are included in Appendix which I appreciate), and I've already raised other concerns about S6 elsewhere.

Clarity: I think the writing can be improved substantially. - Given that the work is largely empirical, the introduction feels unfocused, and does very little in terms of presenting and signposting the central results of the paper. Currently, the reader has very little idea of what to expect in S5/6 until they read till the end of the work. - The experiments themselves are somewhat haphazard, and I would ideally like to see more detail on exactly what the experimental hypotheses are (at the beginning of S5), and why the experiments being run are the right way to test those hypotheses. - I think the experiments could benefit from explaining methodology a little more. As pointed out elsewhere, many of the figures require additional clarification.

Relation to Prior Work: The relationship to prior work was clearly stated.

Reproducibility: Yes

Additional Feedback: I think there is scope for improving the clarity, results and presentation of the paper. - Lines 45-48 suggest that we will see an experiment that validates that distributional shifts under corruption are largely due to the shifts in the first and second moments of the internal activations. However, I don't think this is directly addressed by an experiment in S5/6. While it seems that the BN adaptation can compensate for performance degradation due to these moments, it's not clear that there isn't a difference in higher order moments that is not addressed by this technique. It would be good to address this claim. - Defn 1 seems to be missing some statement about how p_s and p_t are different. - I don't understand Eqn. (3). Aren't p_s and p_t substantially different due to the covariate shift? - It would be nice to see an experiment that quantifies the degree to which the adaptive BN technique can compensate for covariate shift, as the amount of such shift is varied. Concretely, if the IN-C corruptions were sorted in order of the covariate shift induced, does adaptive BN compensate more for the corruptions with larger shift? - Line 149-150 make reference to a full green line with stars: but Figure 1 (i) contains no such line. What is "N best" in Figure 1? It would also be nice to put $n$ on the x-axis of the plot. - In Figure 3 (i), each corruption has multiple points scattered on the plot: what do these correspond to? - I'm confused by Figure 3 (ii). For layer j > i, if W_j < W_i (i.e. there is less divergence at a later layer), does that not indicate that the model has compensated for the distributional shift? How was this figure generated? I don't see any details for this in the Appendix. - For pretraining, how does the distribution of internal activations change? The pretraining section suggests that these models don't require adaptation, so it would be nice to see more analysis on why this is the case. - The theoretical analysis and accompanying empirical study (end of S6) seem like an afterthought. I found this portion largely unclear (see below). - Figure 5 merits more explanation and I found it hard to interpret, nor did I understand its significance. What empirical observation is explained by Fig 5? - Why focus on the W2 distance in Prop 1 rather than the parameter estimation error? - In Prop 1, what is the difference between \hat{\mu}_t and \bar{\mu}_t? - Why would choosing N by minimizing L make sense? Isn't L a lower bound? We would presumably want to minimize U to reduce the divergence between the true statistics and the estimated statistics? - How do you calculate min_N U? Doesn't that require knowledge of the target statistics (mu_t, sigma_t), which is what you're trying to estimate? - Page 2 of the Appendix appears to be incomplete. - The authors clearly made substantial effort in making sure the work is reproducible. I went through the submitted code, as well as the appendix, and I believe that the results should be reproducible in a reasonable amount of time. I encourage the authors to package and release their code for use in future robustness research. -------- Updated (8/20) --------- I've read through the rebuttal. I was mainly concerned by readability issues and quality of presentation, as well as problems with understanding the utility of the theoretical results in the work. Some of these concerns were echoed by other reviewers. I think the authors' did a good job addressing concerns about clarity. I was glad to see that they explicitly wrote out the relationship of experiments to a set of concrete experimental hypotheses. They also updated and have fixed the issues around readability of figures and tables. Some of the other reviewers had valid concerns about generalizing to other datasets, which they made effort to respond to. I'm still somewhat concerned about the presentation and significance of the theoretical results, which remain largely an afterthought. I had specific questions related to the applicability of the results in practice. However, given the space constraints in the rebuttal, the effort made by the authors, and that this is not the central contribution of the work, I'm willing to overlook these issues. As I stated in my review, the simplicity of the work is appealing. Given this, I've updated my score from 5 -> 7 to reflect my updated evaluation of their work. However, whether the paper is accepted or not, I do urge the authors to consider some of the points raised about the theoretical results and communicate them more clearly.


Review 4

Summary and Contributions: The paper suggests to update the batch normalization statistics during evaluation time, in an unsupervised manner, in order to improve robustness metrics on different types of corruptions. This simple adaptation method shows that the mean corruption error in ImageNet-C can be largely reduced, and gives boosts to many published types of neural networks. This suggests that the estimate of the mean corruption error, at least in datasets using ad-hoc distortions such as ImageNet-C, is very likely over-estimated.

Strengths: The paper adopts the proposed simple technique on dozens of pre-trained models on ImageNet and shows that it can greatly reduce the mean corrupted error, when evaluated on ImageNet-C. They also perform experiments with state-of-the-art methods on the same evaluation dataset showing similar improvements. For all these experiments, they provide sufficient details to reproduce the results. Notice that many training details are not mentioned in the paper, but that is ok, since they downloaded the models directly from torchvision. However, I would suggest to add the commit number, or at least the date on which the models were retrieved, just in case there are updates of these models in the future. The authors also made the effort to evaluate this approach on other robustness benchmarks such as ImageNet-A, ImageNet-V2 and ObjectNet, although the results are not that positive in these (see comments later). The paper includes different ablation studies (in addition to the different benchmark datasets), such as pre-training on large datasets, and using alternatives to BatchNorm that try to alleviate the same problem, such as GroupNorm and Fixup Initialization. At least on ImageNet-C, their approach offers better results than using GroupNorm/FixUp, but when using large pre-training datasets, the improvements are not that large.

Weaknesses: The main weakness is that the work seems to highlight (and partially solve) a specific problem of the ImageNet-C benchmark, rather than showing that it is a good approach for training robust neural network-based image classifiers in general. This is shown by the fact that the improvements shown in ImageNet-C are not observed in other benchmarks such as ImageNet-V2, and the more recent ObjectNet benchmark. Given that, as the authors highlight, the distortions in the ImageNet-C benchmark are built ad-hoc, and (perhaps) are not representative of the typical distortions found in more realistic scenarios, one wonders what is the applicability of the proposed approach out of the ImageNet-C benchmark. In addition, the idea of adapting batch normalization statistics is not new in the area. The authors cite four works in which they use the same or similar ideas for other goals (task adaptation, multi-task, ...), see citations [4-7].

Correctness: The empirical methodology is correct, but as mentioned above the results on other benchmarks different from ImageNet-C suggest that the proposed approach only works in this particular (and unrealistic) scenario. In the comparison against GroupNorm and FixUp, the authors only performed experiments on ImageNet-C, while ignoring the other datasets. For instance, on ObjectNet, a R50 using GroupNorm instead of BatchNorm, trained on LSVRC2012, performs much better than the best results reported in this paper: ~58% top-1 error (this paper) vs ~48% top-1 error (https://arxiv.org/pdf/1912.11370.pdf).

Clarity: The paper is generally well written and structured. However, the presentation of some figures could be improved. For instance, figure 4 shows two baseline lines for ObjectNet, but there differences among the two are not stated anywhere. Likewise, the ImageNet-V2 plot shows lines for "Matched", "Threshold" and "Top", but there's no reference to these terms in the text. Also, figure 5 is too small and can be barely read without the use of the magnifier tool in the PDF reader application (not to mention if the paper is printed). It is unclear why some numbers are missing in Table 1. In particular, there are several results missing for Assemble Net. Finally, when referring to the sections in the appendix, it would be preferable to cite the particular subsection, since some sections contain many of those. For instance, the proof (sketch) of Proposition 1 is in Appendix D.8, but it is only referenced as Appendix 8, which contains 11 subsections across 10 pages.

Relation to Prior Work: In the related work section, the paper includes references to several relevant works in robustness and unsupervised domain adaptation, but in some cases it is not explicitly stated what are the differences with respect to this work. For instance, line 252 states: "The idea of adapting activation statistics was originally developed in [44]", but how does it differs from the presented here? The authors also cite alternatives such as GroupNorm and FixUp initialization, although they do so in the experiments, rather than related work section. However, as explained above these alternatives are not fairly compared, since only ImageNet-C is used in this comparison, and good results using these techniques on the other benchmarks are not discussed.

Reproducibility: Yes

Additional Feedback: - Figure 2 shows the IN top 1 error and the IN-C mCE. I am assuming that the top1 error is computed on the eval set of LSVRC2012. It is not clear why this figure is relevant. After adaptation, one would expect the error in the original training distribution (LSVRC2012) to be greater or equal than before the adaptation. So, what's the purpose of the x-axis in this figure? On the contrary, the y-axis does provide useful information for the purposes of this paper. I would suggest to use a bar plot for this figure, showing only IN-C mCE, instead. --------------- Update (2020.08.23) --------------- I appreciate the effort made by the authors in addressing the most important concerns that I raised in my original review. In their feedback, they provide results in additional datasets and explain why their method is not expected to address more complex shifts in data distribution, such as the ObjectNet dataset. In addition, they have presented new results in 15 additional dataset shifts in the ImageNet-R dataset. In their response, the authors also claim to have improved the readability of the manuscript following the reviewers comments. I am still not convinced that the type of data distribution shifts that the proposed approach fixes are the most "common" or most important in "reality". However, I acknowledge that it's hard to define what are the distribution shifts that are seen in "reality", and the rest of reviewers consider that the ones addressed in this paper are important for the community. Given this, I am updating my score from 4 to 6 (i.e. "Marginally above the acceptance threshold").

[Author Response · NeurIPS 2020]

*Overview. We include ImageNet-R results, better models on ImageNet-C and improve our ablation studies.*

We thank the reviewers for their extensive and helpful comments which contributed to improving our manuscript. The reviewers state that the "simplicity of the method is appealing, and it provides a substantial improvement for little extra effort" (R3) and agree on the importance of the considered problem for the NeurIPS community (R1,2). Below, we address main concerns and discuss updated results with more robust models (DeepAugment) and new datasets (ImageNet-R) which appeared in parallel work during the review phase. We also incorporated most of the suggestions regarding figure formatting and formal methods in the camera-ready version.

**R1, R2, R4: Does the proposed method generalize to other datasets?** We already showed gains across the 15 different datasets in the IN-C benchmark (of four different types). We now extend this analysis to 15 new data shifts in ImageNet-R (IN-R; 200 class IN, 30,000 images), another large image dataset with more challenging dataset shifts like art, cartoons, deviantart or graffiti. We observe consistent gains (Table 1) with a new RN50 SoTA of 48.9% when using a batch size of 2048 for adaptation. For the vanilla RN50, we observe performance improvements on IN-R when using a batch size larger than 32 (Fig. 1) almost reaching AugMix performance w/o adaptation for large batch sizes.

T1: ImageNet-R (n=2048), top-1 error.

| Model, adaptation: | base | adapt |
|---|---|---|
| ResNet50 | 63.8 | 59.9 |
| Fixup | 61.2 | — |
| GroupNorm | 65.0 | — |
| SIN | 58.6 | 54.2 |
| ANT | 61.0 | 58.0 |
| ANT+SIN | 53.8 | 52.0 |
| AugMix (AM) | 59.0 | 55.8 |
| DeepAug (DAug) | 57.8 | 52.5 |
| DAug + AM | 53.2 | 48.9 |
| DAug + AM (RNXt101) | **47.9** | **44.0** |

Fig 1: ImageNet-R results

Table 2: New models on IN-C (n=2048), mCE

| Model | base | adapt |
|---|---|---|
| DeepAug | 60.36 | 49.44 |
| DeepAug+AugMix | 53.55 | 45.36 |
| DeepAug+AugMix+RNXt101 | **44.52** | **37.96** |

| T. 3a: ObjectNet evaluation (n = 512), acc | | | T. 3b: Mixed IN-C, err | |
|---|---|---|---|---|
| ResNet50 model | top-1 | top-5 | top-1 | top-5 |
| BatchNorm w/o adapt | 21.85 | 39.09 | 61.08 | 40.81 |
| BatchNorm w/ adapt | 24.04 | 41.15 | 60.87 | 40.31 |
| GroupNorm | **29.18** | **50.24** | 57.25 | 35.97 |
| Fixup | 28.52 | 48.56 | **56.83** | **35.43** |

**Clarifications around novelty & central hypotheses:** Adaptation of BN layers is a well-known method in domain adaptation. Our contribution is to extensively evaluate (and theoretically analyze) its performance on *systematic* dataset shifts in both large and small sample size adaptation scenarios, and to show that a domain adaptation evaluation scenario has the potential to substantially improve over the ad-hoc setting on robustness datasets, making it a strong baseline. Our main hypotheses (**H**) and tests (**T**) (asked by R3) are:

- **H**: *Systematic* dataset shifts yield a mismatch in internal statistics and result in decreased accuracy. **T**: The Wasserstein distance between source and target statistics quantifies the amount of mismatch and is predictive of degradation, especially within a corruption type.
- **H**: Correcting the statistics improves accuracy under distribution shift. **T**: We show consistent, substantial improvements due to BN adaptation across a wide range of models and 17 domains (15 IN-C + IN-R + ON).
- **H**: The observed sample size performance trade-off can be explained by statistical estimation errors (theoretical model) and can be mitigated using a Bayesian approach. **T**: We propose a theoretical model to qualitatively explain the sample size vs. performance degradation trade-off and propose an easy fix for the small sample case.

**Additional Control Experiments (ObjectNet, mixed IN-C)** R4 discussed our negative results on IN-V2 and ON. We want to stress that these results are control experiments, and the observed outcome matches the expectations. BatchNorm adaptation can only mitigate *systematic shifts* in the data distribution, which is unlike the shift in IN-V2 (iid data, or a more complex sampling bias) or ObjectNet (complex distributional shift by random variations in poses, etc).

To stress this point, we perform two additional controls: We evaluate GroupNorm + Fixup on ObjectNet as suggested by R4, which outperform the BN model (T3a). We also randomly sample 50,000 IN-C images across corruptions and severities (3 seeds), destroying the systematic shift. GN+Fixup now also outperforms BN w + w/o adapt (Table 3b).

**Use of exponential moving average instead of a weighted average (R1)** We agree that this is the correct method especially for practitioners, and added a note in the Appendix. Results are indistinguishable from the "full adaptation" results due to the large number of samples in the test set and we can add a short comparison on this to the appendix.

**Manuscript edits** We fixed Figs.1,2,4 according to R2's suggestions; the color code in Fig. 4, IN-V2 was indeed wrong, colors should match in the limit of many samples (adaptation converges to baseline performance). We revised § 1–2 & fixed Def. 1. We revised Fig. 3 and note linear relationships between the Wasserstein distance & accuracy *both before and after* adaptation, highlighting the usefulness to quantify domain shift; we do not observe a relationship between Wasserstein distance and the amount of correction by adaptation (R3) and will add a supplementary figure. We thoroughly revised the appendix and sectioning.

[Meta-Review · NeurIPS 2020]

The authors propose a way to improve the robustness to corruptions by adapting batchnorm statistics on the test set, and show that this improves performance significantly on multiple benchmarks. The reviewers initially raised a few concerns particularly around generalization to other datasets and connections to related work. The authors did a great job of responding to the reviewers' questions including some additional experiments. During the follow-up discussion, the reviewers agreed that the revised experiments satisfactorily address some of the major concerns and some of them increased scores as well. Overall, I think this is a good paper (particularly with the extensive additional results) and I recommend acceptance. I encourage the authors to add the new results (new datasets + ablations) as well as revise the related work section (points raised in the rebuttal + differences from other parallel work that R1 mentioned) in the camera ready version.